



# Potential evaporation at eddy-covariance sites across the globe

Wouter H. Maes[1], Pierre Gentine[2], Niko E.C. Verhoest[1], Diego G. Miralles[1]

1 Laboratory of Hydrology and Water Management, Ghent University, Coupure Links 653, 9000 Gent, Belgium
2 Department of Earth and Environmental Engineering, Columbia University, New York, 10027, USA

*Correspondense to:* Wouter H. Maes (wh.maes@ugent.be)

**Abstract.** Potential evaporation ($E_p$) is a crucial variable for hydrological forecast and in drought monitoring systems. However, multiple interpretations of $E_p$ exist, and these reflect a diverse range of methods to calculate $E_p$. As such, a comparison of the performance of these methods against field observations in different global ecosystems is badly needed. In this study, we used eddy-covariance measurements from 107 sites of the
FLUXNET2015 database, covering 11 different biomes, to parameterize and compare the main $E_p$ methods and uncover their relative performance. For each site, we extracted the days for which ecosystems are unstressed based on both an energy balance approach and on a soil water content approach. The evaporation measurements during these days were used as reference to validate the different methods to estimate $E_p$. Our results indicate that a simple radiation-driven method calibrated per biome consistently performed best, with a mean correlation of 0.93,
an unbiased RMSE of 0.56 mm day$^{-1}$, and a bias of -0.02 mm day$^{-1}$ against in situ measurements of unstressed evaporation. A Priestley and Taylor method, calibrated per biome, performed just slightly worse, yet substantially and consistently better than more complex Penman, Penman-Monteith-based or temperature-based approaches. We show that the poor performance of Penman-Monteith based approaches relates largely to the fact that the unstressed stomatal conductance was assumed constant. Further analysis showed that the biome-specific
parameters required for the simple radiation-driven methods are relatively constant per biome. This makes this simple radiation-driven method calibrated per biome a robust method that can be incorporated into models for improving our understanding of the impact of global warming on future global water use and demand, drought severity and ecosystem productivity.

## 1 Introduction

Since its introduction 70 years ago by C. W. Thornthwaite (1948), the concept of potential evaporation ($E_p$), defined as the amount of water which would evaporate from a surface unconstrained by water availability, has been widely used in different fields. It has been incorporated in numerous hydrological models dedicated to estimate runoff (e.g. Schellekens et al., 2017) or actual evaporation as well as in drought severity indices (Vicente-Serrano et al., 2013;Sheffield et al., 2012). Changes in $E_p$ have been regarded as the main driver of ecosystem
distribution and aridity (Scheff and Frierson, 2013) and are used to estimate the influence of climate change on ecosystems based on climate models projections (e.g. Milly and Dunne, 2016).

However, many different definitions of $E_p$ exist, and consequently many different methods to calculate it. In recent years, there has been an increasing awareness of the impact of the underlying assumptions and caveats of the $E_p$ methods (e.g. Sheffield et al., 2012;Kingston et al., 2009;Seiller and Anctil, 2016;Bai et al., 2016;Guo et al.,
2017;Weiß and Menzel, 2008;Milly and Dunne, 2016). Therefore, a global appraisal of the most appropriate method for assessing $E_p$ of actual ecosystems is badly needed. Current methods disagree on the mere meaning of





this variable, which requires the definition of a reference system (Lhomme 1997). As such, $E_p$ has been defined as either the evaporation which would occur in the given meteorological conditions if water was not limited *(i)* over open water (Shuttleworth, 1993), *(ii)* over a reference crop (usually a short green grass completely shading the ground, and either completely wet (Penman, 1963) or irrigated (Allen et al., 1998)), or *(iii)* over the actual

ecosystem transpiring at a maximal rate (Brutsaert, 1982;Granger, 1989).

A second source of disagreement on the definition of $E_p$ relates to the extent of the reference system and the influence of the reference system on the meteorological conditions. An idealized extensive ecosystem evaporating at maximum rate can be expected to raise air humidity until the vapour pressure deficit tends to zero. In that case, evaporation is driven by a radiation (energy supply) component only. In addition, considering such an extensive

area also avoids problems with advection and entrainment flows. Consequently, such an extensive reference system was used to define $E_p$ by many authors (Penman, 1963;Priestley and Taylor, 1972;Brutsaert, 1982;Shuttleworth, 1993). Others opined that the meteorological conditions should not depend on the extent of the reference system, by making this reference system infinitesimally small (Morton, 1983;Gentine et al., 2011b;Pettijohn and Salvucci, 2009). Upon all this controversy, the net radiation of the reference system remains

a point of discussion: some argue that the reference system should have the same net radiance as the actual system (e.g. Rind et al., 1990;Crago and Crowley, 2005;Granger, 1989). Yet, this is inherently inconsistent as the surface skin temperature reflects the surface energy partitioning. Thus, a system transpiring at potential rate has a lower skin temperature (Maes and Steppe, 2012) and correspondingly a lower net radiation (e.g. Lhomme, 1997;Lhomme and Guilioni, 2006).

As can be concluded from the above discussion, it is nearly impossible to define a correct and universally accepted definition of $E_p$, and the most appropriate definition should remain tied to the specific interest and application. Nonetheless, as different applications make use of different $E_p$ methods, a good knowledge of the implications of different $E_p$ definitions is required (Fisher et al., 2011). If considering terrestrial ecosystems, the open water reference system seems less informative of the available energy and the aerodynamic properties of the ecosystem

(Shuttleworth, 1993;Lhomme, 1997). Considering that the well-watered crop system only takes climate forcing conditions into account and does not require information on land cover, it has become the universally most often used reference system (Lhomme, 1997) and is traditionally the preferred system when looking at the global water demand or drought severity (Dai, 2011). When the reference system is considered the actual ecosystem transpiring at maximal rate, both climate forcing conditions and ecosystem properties need to be taken into account. This is

the preferred system when $E_p$ is used as a means to estimate actual evaporation ($E_a$). This is commonly done by applying a multiplicative stress function, $\beta$, varying between 0 and 1, such that $E_a = \beta E_p$ (Barton, 1979). This sort of stress functions are used by evaporation retrieval models based on remote sensing or weather station data at regional or global scales (e.g. Fisher et al., 2008;Miralles et al., 2011b;Martens et al., 2017;Mu et al., 2007), and can be considered analogous to the $\beta$ factor used in most land surface models to incorporate the effect of soil

moisture in the estimation of surface turbulent fluxes (Powell et al., 2013). This definition of $E_p$ based on the actual ecosystem as reference system is also preferred for global streamflow and runoff studies based on the Budyko approach.

When comparing and evaluating different $E_p$ methods, studies using a modelling approach have either *(i)* compared the performance of different $E_p$ methods in hydrological models (Oudin et al., 2005a;Seiller and Anctil,

2016;Xu and Singh, 2002;Kay and Davies, 2008) or *(ii)* in climate models (Milly and Dunne, 2016;Weiß and





Menzel, 2008;Lofgren et al., 2011), or (*iii*) used the Penman-Monteith method as the benchmark to test different alternative $E_p$ formulations that require less input data (e.g. Chen et al., 2005;Sentelhas et al., 2010). Although these studies have their merits, it is clear that approaches in which the $E_p$ methods are evaluated based on empirical data of actual measurements of evaporation should be preferred. Unfortunately, such approaches are hampered by
limited data availability (Weiß and Menzel, 2008). Lysimeter data, probably the most precise evaporation measurements available, have been used (e.g. Pereira and Pruitt, 2004;Katerji and Rana, 2011;Yoder et al., 2005), but measurements are scarce and difficult to upscale to larger ecosystems. Pan evaporation data, available in larger volumes and at larger scales, have also been used (Donohue et al., 2010;Zhou et al., 2006;McVicar et al., 2012) but provide a proxy of open-water evaporation, rather than actual ecosystem potential evaporation, and also exhibit
biases related to the location, shape and composition of the instrument (Pettijohn and Salvucci, 2009). Eddy covariance measurements, finally, are an attractive alternative, but have so far been used in a relatively limited number of studies focusing on local or regional scale only (Douglas et al., 2009;Li et al., 2016;Jacobs et al., 2004;Sumner and Jacobs, 2005).

The purpose of this article is to identify the best method to estimate $E_p$ in different ecosystems across the globe.
Based on the above review, $E_p$ is defined using the actual ecosystem evaporating at maximal rate as reference system, so $E_p$ refers to the actual atmospheric demand for water experienced by the ecosystem. We used the most recent and complete eddy-covariance database available, i.e. the FLUXNET2015 archive (http://fluxnet.fluxdata.org/). The most frequently-adopted $E_p$ methods are applied based on standard parameterizations as well as calibrated parameters per biome, and inter-compared in order to gain insights into
the most suitable means to estimate $E_p$ in global models.

## 2. Material and Methods

### 2.1. Selection of $E_p$ methods

Methods to calculate $E_p$ can be categorized based on the amount and type of input data required. In this overview, we will only discuss the most frequently used or new methods that will be evaluated in our study – the readers are
referred to Oudin et al. (2005a) or Seiller and Anctil (2016) for more extensive overviews.

**Methods based on radiation, temperature, wind speed and vapour pressure deficit**

The well-known Penman-Monteith equation (Monteith, 1965) expresses latent heat flux $\lambda E_a$ (W m$^{-2}$) as:

$$\lambda E_a = \frac{s\,(R_n - G) + \frac{\rho_a\,c_p VPD}{r_{aH}}}{s + \gamma + \gamma\frac{r_c}{r_{aH}}} = \frac{s\,(R_n - G) + \frac{\rho_a\,c_p VPD}{r_{aH}}}{s + \gamma + \frac{\gamma}{g_c\,r_{aH}}} \qquad (1)$$

With $\lambda$ the latent heat of vaporisation (J kg$^{-1}$), $E_a$ the actual evaporation (kg m$^{-2}$ s$^{-1}$), $s$ the slope of the Clausius-Clapeyron curve relating air temperature with the saturation vapour pressure (Pa K$^{-1}$), $R_n$ the net radiation (W m$^{-2}$), G the ground heat flux (W m$^{-2}$), $\rho_a$ the air density (kg m$^{-3}$), $\gamma$ the psychrometric constant (Pa K$^{-1}$), $c_p$ the specific heat capacity of the air (J kg$^{-1}$ K$^{-1}$), VPD the vapour pressure deficit (Pa), $r_{aH}$ the resistance of heat transfer to air (s m$^{-1}$), $r_c$ the canopy resistance of water transfer (s m$^{-1}$) (middle equation) and $g_c$ the canopy conductance to water transfer (m s$^{-1}$; $g_c = r_c^{-1}$). While $\lambda$, $c_p$, $s$ and $\gamma$ are air temperature-dependent, $r_{aH}$ is a complex function of wind speed, vegetation characteristics and the stability of the lower atmosphere (see Section 2.3). In most methods to
estimate $E_a$ or $E_p$, $r_{aH}$ is considered as a simple function of wind speed.





The Penman-Monteith equation can be used for calculating $E_p$ by adjusting $r_c$ to its minimum value (the value under stressed conditions). If the reference system is the actual system, or a reference crop transpiring at maximal rate, $r_c$ is usually considered as a fixed, constant value larger than zero. In this study, both a universal, fixed value of $r_c$ (or $g_c$) for reference crops as a biome-specific constant value will be used. If a wet canopy is considered, $r_c=0$ and Eq. (1) collapses to:

$$\lambda E_p = \frac{s\,(R_n - G) + \frac{\rho_a\,c_p VPD}{r_{aH}}}{s + \gamma} \tag{2}$$

Eq. (2) is often referred to as the Penman (1948) method and can be re-arranged to yield $\lambda E_p = \frac{s\,(R_n - G)}{s + \gamma} + \frac{\rho_a\,c_p VPD}{(s+\gamma)\,r_{aH}}$, showing that $E_p$ can be driven by a radiative (left) or an aerodynamic forcing (Brutsaert and Stricker, 1979).

**Methods based on radiation and temperature**

In case the reference system is considered an idealized extensive area, or when radiation is much greater than atmospheric vapour deficit, the aerodynamic component of Eq. (2) tends to 0 and the whole equation collapses to $\lambda E_p = \frac{s\,(R_n - G)}{s + \gamma}$, commonly referred to as equilibrium evaporation (Slatyer and McIlroy, 1961). Priestley and Taylor (1972) analysed time series of open water and water-saturated crops and grasslands and found that the evaporation over these surfaces closely matched the equilibrium evaporation corrected by a multiplication factor $\alpha_{PT}$:

$$\lambda E_p = \alpha_{PT} \frac{s\,(R_n - G)}{s + \gamma} \tag{3}$$

This formulation is known as the Priestley and Taylor equation, and usually a value of $\alpha_{PT}=1.26$ is adopted, as estimated by Priestley and Taylor (1972) in their original experiments. In this study, we will also include a vegetation-specific value. Since this method does not require wind speed or VPD as input, it is one of the $E_p$ methods most widely used in hydrological, remote sensing and drought models.

**Methods based on radiation**

Other studies such as Lofgren et al. (2011), or the more recent Milly and Dunne (2016), further simplified Eq. (3) to:

$$\lambda E_p = \alpha_{MD}\,(R_n - G) \tag{4}$$

In the case of Milly and Dunne (2016) this equation was applied to climate model output based on a value of $\alpha_{MD}=0.8$. The above Eq. (4) can be easily related to the surface energy balance of the ecosystem, which is given by $R_n - G = \lambda E + H$, with H the sensible heat flux (W m$^{-2}$). On a daily scale, $(R_n - G)$ expresses the total amount of energy available for evaporation, and the fraction of this energy that is actually used for evaporation is typically referred to evaporative fraction, or $EF = \frac{\lambda E_a}{(H + \lambda E_a)} = \frac{\lambda E_a}{(R_n - G)}$. From Eq. (4), it follows that the parameter $\alpha_{MD}$ can be interpreted as the EF of the unstressed ecosystem. In this study, we will test both the general value of $\alpha_{MD}=0.8$ and a biome-specific value.





**Methods based on temperature**

Of the many empirical methods to estimate $E_p$, temperature-based methods have been most commonly used because of the availability of reliable air temperature data. For an overview of these methods, we refer to Oudin et al. (2005a). In this study, two methods are included. Pereira and Pruitt (2004) formulated a daily version of the well-known Thornthwaite (1948) equation:

$$T_{eff} < 0 \qquad \lambda E_p = 0 \tag{5a}$$

$$0 < T_{eff} < 26 \qquad \lambda E_p = \alpha_{Th} \left( \frac{10\, T_{eff}}{I} \right)^b \left( \frac{N}{360} \right) \tag{5b}$$

$$26 < T_{eff} \qquad \lambda E_p = -c + d\, T_{eff} - e\, T_{eff}{}^2 \tag{5c}$$

with $T_{eff}$ being the effective temperature, based on maximum and minimal temperatures (see further, Section 2.5), $\alpha_{Th}$ an empirical parameter (see below), I the yearly sum of $(T_{a\_mean}/5)^{1.514}$ , with $T_{a\_mean}$ the mean air temperature for each month, $N$ the number of daylight hours, $b$ a parameter depending on I and $c$, $d$ and $e$ empirical constants (see further , Section 2.5). The general value of $\alpha_{Th} = 16$ is often adopted; in this study, we will also calculate and apply a biome-specific value.

The second temperature-based method is the one proposed by Oudin et al. (2005a), after comparing 27 physically-based and empirical methods with runoff data from 308 catchments:

$$T_a < 5 \qquad \lambda E_p = 0 \tag{6a}$$

$$T_a > 5 \qquad \lambda E_p = \frac{R_e}{\rho_a} \frac{(T_a - 5)}{\alpha_{Ou}} \tag{6b}$$

with $T_a$ being air temperature (°C), and $R_e$ top-of-atmosphere radiation (MJ m$^{-2}$ day$^{-1}$), depending on latitude and Julian day. Oudin et al. (2005a) suggested to use $\alpha_{Ou} = 100$. This value will be used, next to a biome-specific value. A detailed description of the calibration of all $E_p$ methods is given in Section 2.5.

**2.2. FLUXNET2015 Database**

The Tier2 FLUXNET2015 database based on half-hourly or hourly measurements from eddy-covariance sites is used to evaluate the estimates of $E_p$ (http://fluxnet.fluxdata.org/data/fluxnet2015-dataset/). Sites lacking at least one of the basic measurements required for our analysis (i.e. $R_n$, G, $\lambda E_a$, H, wind speed ($u$), friction velocity ($u^*$), $T_a$ and relative humidity (RH) or VPD) were not further considered. For latent heat flux, we used the data corrected by energy balance closure (Michel et al., 2016). For $R_n$ and the main fluxes (G, H, $\lambda E_a$), medium and poor gap-filled data were masked out according to the information provided by FLUXNET. As no quality flag was available for $R_n$ measurements, the quality flag of the shortwave incoming radiation was used instead. All negative values for H or $\lambda E_a$ were masked out, as these relate to periods of interception loss and condensation. Similarly, all negative values of $R_n$ were masked out. Finally, sub-daily measurements were aggregated to daytime composites based on a threshold of 5 W m$^{-2}$ of top-of-atmosphere incoming shortwave radiation and the first and last (half-) hours of the day were excluded from these aggregates; if top-of-atmosphere radiation was not available, surface shortwave incoming radiation was used instead. Based on these daytime values, the daytime means of $s$, γ, $\rho_a$ were calculated using the parameterisation procedure described by Allen et al. (1998). We used air temperature to calculate $s$. Only days in which more than 30% of the data were measured directly were retained, and days with





rainfall (between midnight and sunset) were removed from the analyses to avoid the effects of rainfall interception. Only sites with at least 80 retained days were used for the further analysis. The global distribution of the final selection of sites is shown in Fig. 1 and detailed information about these sites is provided in Table S1 of the Supporting Information. The IGBP-classification was used to assign a biome to each tower.

*(Insert Figure 1)*

### 2.3. Calculation of conductance parameters

Estimates of $r_{aH}$ are required by the Penman and Penman-Monteith equations. The resistance of heat transfer to air, $r_{aH}$, was calculated as:

$$r_{aH} = \frac{u}{u_*^2} + \frac{1}{k\,u_*}\left[\ln\left(\frac{z_{0m}}{z_{0h}}\right) + \Psi_m\left(\frac{z-d}{L}\right) - \Psi_m\left(\frac{z_{0h}}{L}\right) - \Psi_h\left(\frac{z-d}{L}\right) + \Psi_h\left(\frac{z_{0h}}{L}\right)\right] \tag{7}$$

in which $k=0.41$ is the von Karman constant, $z$ the (wind) sensor height (m), $d$ the displacement height (m), $z_{0m}$ and $z_{0h}$ the roughness lengths for momentum and sensible heat transfer (m), respectively, L the Obukhov length (m), and $\Psi_m(X)$ and $\Psi_h(X)$ the Businger-Dyer stability functions for momentum and heat for the variable X, respectively. These were calculated based on the equations given by Garratt (1992) and Li et al. (2017) for stable, neutral and unstable conditions. Note that in neutral and stable conditions, $\Psi_m(X) = \Psi_h(X)$ and that $\Psi_m\left(\frac{z-d}{L}\right) -$

$\Psi_m\left(\frac{z_{0h}}{L}\right) - \Psi_h\left(\frac{z-d}{L}\right) + \Psi_h\left(\frac{z_{0h}}{L}\right) = 0$. This is not the case for unstable conditions, which mostly prevail during the daytime. Daytime averages of all variables were used as input in Eq. (7).

The sensor height $z$ was collected individually for each tower through online and literature research, or personal communication with the towers' P.I. The Monin-Obukhov length L was calculated as (Li et al., 2017):

$$L = \frac{-\,u_*^3\,\rho\,T_a\,(1 + 0.61\,q_a)\,c_p}{k\,g\,H} \tag{8}$$

with $q_a$ being the specific humidity (kg kg$^{-1}$) and $g=9.81$ m s$^{-2}$ the gravitational acceleration.

The displacement height $d$ and the roughness length for momentum flux $z_{0m}$ were estimated as a function of the canopy vegetation height (VH), as $d=0.66$ VH and $z_{0m}=0.1$ VH (Brutsaert, 1982). The VH was estimated from the flux tower measurements using the approach of Pennypacker and Baldocchi (2016):

$$VH = \frac{z}{0.66 + 0.1\exp\left(\frac{k\,u}{u_*}\right)} \tag{9}$$

This equation was applied to the full (half-)hourly database and only when conditions were near-neutral ($|z/L| <$ 0.01) and when friction velocities were lower than one standard deviation below the mean of $u_*$ at each site. The

25 daily VH was then aggregated by averaging out the half-hourly estimates to daily values, excluding the 20% outliers of the data, and then calculating a 30-day window moving average on the dataset, again excluding 20% of the data. This gave robust results for all sites. When not enough (half-)hourly vegetation height observations (<160) were available, the site was excluded from the analysis. An example of VH temporal development for a specific site is given in Fig. 2a.

The Stanton number (defined as $kB^{-1} = \ln(z_{0M}/z_{0H})$) was calculated by assuming that the surface aerodynamic temperature $T_0$ (defined by $H = \rho_a\,c_p\,\frac{(T_0 - T_a)}{r_{aH}}$) is equal to the radiative surface temperature $T_s$ derived from the





longwave fluxes (Li et al., 2017). Then, through an iterative approach, an optimal value of $z_{0H}$ was obtained, using the following equations for $T_0$ (Garratt, 1992) and $T_s$ (Maes and Steppe, 2012):

$$T_0 = T_a + \left(\frac{H}{k\, u_*\, \rho_a\, c_p}\right) \left[\ln\left(\frac{z-d}{z_{0h}}\right) - \Psi_h\left(\frac{z-d}{L}\right) + \Psi_h\left(\frac{z_{0h}}{L}\right)\right] \tag{10}$$

$$T_s = \sqrt[4]{\frac{LW_{out} - (1-\varepsilon)\, LW_{in}}{\sigma\, \varepsilon}} \tag{11}$$

with $\sigma$ the Stefan-Boltzmann constant and $\varepsilon$ the surface emissivity (see further). The (half-)hourly data were used for this calculation. Following the approach of Li et al. (2017), only summertime data were used and only those

5 measurements when H was larger than 20 W m$^{-2}$ and were $u_*$ was larger than 0.01 m s$^{-1}$. Summertime was defined as those months in which the maximal daily value for H is at least 85% of the maximum value for H for the time series at the tower (with the maximum value derived as the 98[th] percentile, to avoid influences from outliers). In addition, (half-)hourly observations with counter-gradient heat fluxes were excluded from the analysis. For each observation, the difference between $T_0$ and $T_s$ was minimized by optimizing $z_{0H}$. Then, the $kB^{-1}$ was calculated at

10 each site based on its relation with the observed Reynolds number (Re) by fitting the following function, based on the work by Li et al. (2017):

$$k\,B^{-1} = a_0 + a_1\, Re^{a_2} \tag{12}$$

Note that Eq. (11) requires a value for $\varepsilon$, which is often assumed to be equal to 0.98 for all sites (e.g. Li et al., 2017;Rigden and Salvucci, 2015). Under the assumption that $T_0 = T_s$, $\varepsilon$ can also be calculated separately per site. If H=0, it follows that $T_0 = T_a$ and from Eq. (11),

$$\varepsilon = \frac{LW_{out} - LW_{in}}{\sigma\, T_a^4 - LW_{in}} \tag{13}$$

Here, $\varepsilon$ was calculated for each site using (half-)hourly data, selecting those measurements where H was close to 0 (-2<H<2 Wm$^{-2}$) and excluding rainy days as well as measurements in which the albedo (calculated as SW$_{out}$/SW$_{in}$) was above 0.4, to avoid measurements of snow or ice. Negative estimates of $\varepsilon$ were filtered out, and the $\varepsilon$ of the site was calculated as the mean excluding the outlying 20% of the data. Equation 3 was applied both with a fixed $\varepsilon$ of 0.98 and with the observed $\varepsilon$, and the equation with the lowest RMSE for Eq. (12) was retained.

An example of such a function between $kB^{-1}$ and Re is shown in Fig. 2b.

*(Insert Figure 2)*

Finally, the canopy resistance $r_c$ (s m$^{-1}$) was calculated as the residual from the Penman-Monteith equation as:

$$r_c = \frac{s\,(R_n - G)\, r_{aH} + \rho_a\, c_p VPD}{\gamma\, \lambda E_a} - \frac{(s + \gamma)\, r_{aH}}{\gamma} \tag{14}$$

We converted the $r_c$ estimates to canopy conductance $g_c$ (mm s$^{-1}$) using $g_c = 1000\, r_c^{-1}$ and will stick to $g_c$ for the remainder of the document Note that the approach of calculating $kB^{-1}$ directly requires a separate measurement of LW$_{in}$ and LW$_{out}$, which was only available in 95 of the 107 selected sites. For the remaining sites, an alternative approach was used to calculate $kB^{-1}$ consisting of estimating $g_c$ with different parameterisations for $kB^{-1}$ and selecting the method with highest R² between $g_c$ and $\lambda E_a$ (see Supporting Information).



### 2.4. Selection of unstressed days

To identify a subset of measurements per eddy-covariance site in which the ecosystem was not undergoing any stress we included two different approaches and provided the results for both methods. A first approach was based on soil moisture levels. For those sites where soil moisture levels were available, the maximal soil moisture level

5 for each site was determined as the 98[th] percentile of all soil moisture measurements. We split up the dataset of each tower in 5 classes, according to the 20[th] percentiles of evaporation, in order to cover unstressed evaporation during all seasons. Of each class, we selected those days with the highest 5% soil moisture levels, but only if these selected days had a soil moisture level above 75% of the maximum soil moisture level.

As soil moisture data were not available for a large number of sites and using soil moisture data does not exclude

10 days in which the functioning of the ecosystems has been affected by other kinds of biotic or abiotic stresses, a second approach for defining unstressed days was additionally applied, using an energy balance criterion. We calculated the EF from the daytime $\lambda E_a$ and H values, and considered it as a direct proxy for evaporative stress (Maes et al., 2011;Gentine et al., 2007;Gentine et al., 2011a). The underlying hypothesis is that under unstressed conditions, a larger fraction of the available energy is used to evaporate. This approach is similar to the one used

in other $E_p$ studies on eddy-covariance or lysimeter data, in which the Bowen ratio (e.g. Douglas et al., 2009) or the ratio of $\lambda E_a/(SW_{in} + LW_{in})$ (Pereira and Pruitt, 2004) are used to define unstressed days. The unstressed record was comprised of all days with EF>95[th] percentile threshold for each particular site, or, if less than 15 days fulfilled this criterium, the 15 days with the highest EF. Consequently, we assume that at each site during at least 5% of the days the conditions are such that evaporation is unstressed and $E_a$ reflects $E_p$. The measured actual evaporation

from the identified unstressed days by either method is further referred to as $E_{unstr}$ (mm day$^{-1}$) and used as reference data to evaluate the different $E_p$ methods.

### 2.5. Calculation and calibration of the different $E_p$ methods

An overview of the different methods to calculate $E_p$ is given in Table 1. If possible, a reference crop, standard and biome-specific version of each method is calculated. The reference crop version calculates $E_p$ for the reference

crop, with the estimated outgoing radiation and other properties of the reference crop. The standard version considers the radiation and other properties of the actual crop but uses the non-biome-specific parameters of the reference crop. The biome-specific version considers the radiation and other properties of the actual crop and applies a calibration of the key parameter (Table 1) of each method. This calibration values per biome is based on the mean value of this key parameter of the unstressed dataset for each tower, averaged out per biome.

To estimate the radiation and crop properties of the reference crop versions, the equations described by Allen et al. (1998) were used and G was considered to be 0. $R_n$ was calculated as:

$$R_n=SW_{in}(1-\alpha_{ref}) + LW^*$$ (15)

with $\alpha_{ref}$=0.23 (Allen et al., 1998) and $LW^*$ being the net longwave radiation, calculated after Allen et al. (1998; Eq. (39), Chapter 3) based on minimal and maximal daily temperature, actual vapour pressure and relative shortwave radiation.

In the case of the reference crop version of the Penman-Monteith equation (Eq. (1)), the FAO-56 (Food and Agricultural Organization) method was used as described by Allen et al. (1998), with $g_{c\_ref}$ fixed as 14.49 mm s$^{-1}$ (corresponding with $r_{c\_ref}$= 69 s m$^{-1}$) and using Eq. (15) to calculate $R_n$. The standard version of the Penman-Monteith equation used observed ($R_n$, G, VPD) and calculated ($s$, $\gamma$, $\rho_a$, $r_{aH}$) daytime values as described in Section



2.2. in Eq. (1), and assumed $g_{c\_ref}$ = 14.49 mm s$^{-1}$. The biome-specific version was calculated with the same data but used a biome-dependent value of $g_c$. First, for each individual site, the unstressed $g_c$ was calculated as the mean of the $g_c$ values of the unstressed record (see Section 2.4). The mean value per biome $g_{c\_ref}$ was then calculated from these unstressed $g_c$ values. Regarding the Penman method (Eq. (2)), the reference crop and standard versions were calculated using the same input data as for the Penman-Monteith methods, and as $g_c = \infty$

($r_c$=0), no biome-specific version was calculated.

The reference crop version of the Priestley and Taylor method is calculated from Eq. (3) with $R_n$ from Eq. 15, $s$ and $\gamma$ from the FAO-56 calculations, and with $\alpha_{PT}$ = 1.26. The standard version uses the same value for $\alpha_{PT}$ but the observed daytime values for $R_n$ and G. The biome-specific version followed a calibration of $\alpha_{PT}$ similar to the

$g_{c\_ref}$ calculation. For each site, the unstressed $\alpha_{PT}$ was calculated as the average $\alpha_{PT}$, obtained by solving Eq. (3) for $\alpha_{PT}$, of the unstressed dataset. Finally, the mean per biome was calculated and used in the $E_p$ estimation. Regarding the method by Milly and Dunne (2017) (Eq. (4)), the reference crop, standard and biome-specific calculation were calculated accordingly, with $R_n$ from Eq. (15) for the reference crop version, $\alpha_{MD}$=0.8 for the reference crop and standard version, and a calibrated $\alpha_{MD}$ per biome type for the biome-specific version.

For Thornthwaite's and Oudin's methods (Eq. (5)), only a standard and a biome-specific version were calculated. The standard version uses $\alpha_{Th}$ = 16. In the biome-specific version, this parameter was again calculated per site as the mean value of the unstressed records (e.g. Xu and Singh, 2001;Bautista et al., 2009) and then averaged per biome. The effective temperature $T_{eff}$ was calculated as $T_{eff} = 0.36\ (3T_{max} - T_{min})$ (Camargo et al., 1999). The parameter $b$ was calculated as $b = (6.75\ 10^{-7}I^3)- (7.71\ 10^{-7}I^2) + 0.0179I + 0.492$ and the parameters $c$, $d$

and $e$ in Eq. (5c) are -415.85, 32.24 and 0.43, respectively. Finally, for Oudin's temperature-based method, $\alpha_{Ou}$ = 100 was taken for the standard version (Eq. (6)). In the biome-specific version, this value was recalculated by calculating $\alpha_{Ou}$ for the unstressed records through Eq. (6), calculating the mean $\alpha_{Ou}$ per site and finally the biome-dependent $\alpha_{Ou}$. Altogether, this exercise yielded a total of 15 different methods to estimate $E_p$ whose specificities are documented in Table 1.

*(Insert Table 1)*

## 3. Results

### 3.1. Key parameters per biome

We first focus on the parameter estimates of the unstressed record of the energy balance criterion (Section 2.4).

Of the full dataset, 107 flux sites meet all the selection criteria (At least 80 days without rainfall and good quality measurements of radiation and main fluxes and at least 160 vegetation height observations, see Sections 2.2 and 2.3). Despite considerable variation within each biome, statistically significant differences are observed among biomes for all of the key parameters of the unstressed records (see Sect. 2.3), although these differences are only marginally significant in the case of $g_{c\_ref}$ ($p$=0.017 – see Table 2). Overall, croplands (CRO) are characterised by

a higher measured $E_{unstr}$, translated in the highest $g_{c\_ref}$, $\alpha_{PT}$, $\alpha_{MD}$, $\alpha_{Th}$ and the lowest $\alpha_{Ou}$ of all biomes. Deciduous broadleaf forest (DBF) and evergreen broadleaf forest (EBF) also have high $g_{c\_ref}$, $\alpha_{PT}$, $\alpha_{MD}$, $\alpha_{Th}$ but low $\alpha_{Ou}$,





while savannah ecosystems (Woody savannah (WSA), Savannah (SAV) and Open shrublands (OSH)) are characterised by lower $E_{unstr}$ and lower $g_c$, $\alpha_{PT}$, $\alpha_{MD}$, $\alpha_{Th}$ and higher $\alpha_{Ou}$.

Only five sites (DE-KLI and IT-BCi, croplands; CA-SF3, OSH; AU-Rig, grassland (GRA) and AU-Wac, evergreen broadleaf forest) have mean values of $\alpha_{PT}$ higher than 1.26 (Table 2). In contrast, 27 sites, of which 9 croplands, have a mean value of $\alpha_{MD}$ above 0.80 and 42 sites have mean $g_{c\_ref}$ above 14.49 mm s$^{-1}$. Wetlands (WET) are located in tropical, temperate as well as in arctic regions, explaining the large standard deviation of $\alpha_{PT}$ and $\alpha_{RB}$ (Table 2).

Next, the effect of the climate forcing variables on $E_{unstr}$ and on the key parameters $g_{c\_ref}$, $\alpha_{PT}$ and $\alpha_{MD}$ is investigated. Fig. 3 gives the distribution of the correlations between the climate forcing variables and $E_{unstr}$, $g_{c\_ref}$, $\alpha_{PT}$ and $\alpha_{MD}$ of the unstressed records at each site. We did not include $\alpha_{Th}$ or $\alpha_{Ou}$ because temperature-based methods did not perform well (see next Section). $E_{unstr}$ is strongly positively correlated with $R_n$, $T_a$ and VPD, but less with $u$ (Fig. 3a, Table 3).

Across all sites, the correlation between $g_{c\_ref}$ and the forcing variables was not significantly different from zero against any climate variable. Nevertheless, $g_{c\_ref}$ was significantly negatively correlated with $T_{air}$ and with VPD in 40 and 45% of the flux tower sites, respectively (Table 3, Fig. 3b). The two parameters $\alpha_{PT}$ and particularly $\alpha_{MD}$ correlated much less to any climate variable across all sites (Table 3b). Consequently, the distributions of the correlations of the climate forcing variables with $\alpha_{MD}$ are peaking around zero (Fig. 3c): $\alpha_{MD}$ is hardly influenced by $R_n$, and is overall not dependent on $u$, $T_a$, [$CO_2$], or VPD in most sites (Fig. 3c, Table 3).

*(Insert Table 2)*
*(Insert Table 3)*
*(Insert Figure 3)*

### 3.2. Evaluation of different $E_p$ methods

We first list the results of the analysis using the energy balance criterion for selecting the unstressed records (Section 2.4). The scatterplots of measured $E_{unstr}$ versus estimated $E_p$ based on the 15 different methods are shown in Fig. 4 for a total of six sites belonging to different biomes. Despite the overall skill shown by the different $E_p$ methods, considerable differences can be appreciated. In general, reference crop methods (PM$_r$, Pe$_r$, PT$_r$, MD$_r$) overestimate $E_{unstr}$ and only two methods, MD$_B$ and PT$_B$, match the measured $E_{unstr}$ closely.

*(Insert Figure 4)*

Table 4 gives the mean correlation per biome for each method. The results are very consistent and reveal that the highest correlations for nearly all biomes are obtained with the standard and biome-specific radiation-based method (MD$_s$ and MD$_b$), closely followed by the standard and biome-dependent Priestley and Taylor method (PT$_s$ and PT$_b$). Temperature-based methods have the lowest overall mean correlation as well as lower mean correlations per biome. Note that the correlations are the same for the standard and biome-specific version in the case of Priestley and Taylor (PT$_s$ and PT$_b$), Milly and Dunne (MD$_s$ and MD$_b$) and Oudin (Ou$_s$ and Ou$_b$) (Table 4) – this





is to be expected, as the only difference between the standard and biome-specific version of these methods is their key parameters ($\alpha_{PT}$, $\alpha_{MD}$, $\alpha_{Ou}$) which are multiplicative (see Eqs. 3, 4 and 6). Differences are however reflected in their unbiased Root Mean Square Error (unRMSE) and mean bias– see Tables 5 and 6. The biome-specific versions of the radiation-based method ($MD_b$) and of the Priestley and Taylor method ($PT_b$) have consistently the

lowest unRMSE for all biomes. Though the difference between these two methods is small, $MD_b$ is performing slightly better. The standard Penman method ($Pe_s$) has the highest unRMSE. All reference crop methods ($PM_r$, $Pe_r$, $PT_r$, $MD_r$) have mean unRMSE above one, and the temperature-based methods ($Th_s$, $Ou_s$, $Th_b$, $Ou_b$) also have relatively high unRMSE. Finally, bias estimates are given in Table 6. Again, $MD_b$ is overall the best performing method (mean bias closest to 0), closely followed by the $PT_b$ method. Both methods have consistent low bias

among all biomes, except for wetlands. Most reference crop methods ($PM_r$, $Pe_r$, $PT_r$, $MD_r$) as well as $Pe_s$ overestimate $E_p$ in all biomes.

*(Insert Table 4)*

*(Insert Table 5)*

*(Insert Table 6)*

The use of soil moisture content as criterion to select unstressed days (see Sect. 2.4) is explored. In total, 62 sites have soil water content data and meet the other selection criteria documented in Section 2.2. The results of this analysis are given in Tables S2-S4 of the supporting section. To allow for a fair comparison, the same statistics

have also been computed for just the same 62 tower sites with the energy balance-criterion (Tables S5-S7). Using the soil moisture criterion, the correlations are overall lower and the results of the mean correlation, unRMSE and biases are less consistent. However, the overall performance ranking of the different models remains similar: $PT_b$ is the best performing method with overall the highest mean correlation (R=0.84) and the lowest unRMSE (0.78 mm/day) and the bias closest to zero (-0.04), closely followed by the $MD_b$ method, with R=0.81, unRMSE=0.89

and a mean bias of -0.12.

So far, all flux sites were used to calibrate the key parameters (Table 2) and those same sites were also used for the evaluation of the different methods. This was done to maximise the sample size; however, to test for possible overfitting, we also repeat the analysis after separating calibration and validation samples. For each biome, two-thirds of the sites were randomly selected as calibration sites, and one third as validation sites. The key parameters

were then calculated from the calibration subset, and applied to estimate $E_p$ of the biome-specific methods of the validation subset. This procedure was repeated 100 times and the mean correlation, unbiased RMSE and bias per biome were calculated. These results are provided in Tables S8-S10, and show no substantial differences in overall correlation, unRMSE and bias of each method.

**4 Discussion**

**4.1 Comparison of criteria to define unstressed days**

We prioritised the energy balance over the soil moisture criterion to select unstressed days, because it can be applied to sites without soil moisture measurements and because it implicitly allowed the exclusion of days in which the ecosystem is stressed for reasons other than soil moisture availability (e.g. insect plagues, phenological





leaf-out, fires, heat and atmospheric dryness stress, nutrient limitations). In addition, surface soil moisture can be a poor indicator of water stress, as rooting depth can vary and is not accurately measured, and different plants may exhibit various strategies and responses to water stress (Miralles et al., 2011a;Douglas et al., 2009;Powell et al., 2006;Martínez-Vilalta et al., 2014).

This is confirmed by our results: sampling unstressed days based on the energy balance-based criterion resulted in higher correlations (Table S5 vs Table S2) between $E_p$ and $E_{unstr}$ for all methods and in lower unRMSE, with the exception of the temperature-based methods (Table S6 vs Table S3). However, the soil moisture criterion provides an independent check of the results and confirms the superior performance of the $PT_b$ and $MD_b$ methods. In the following sections, discussions are therefore focused primarily on the results of the energy balance method.

**4.2 Estimation of key ecosystem parameters**

The biome-specific values of the key parameters in Table 2 were within the range of values used in reference crop and standard application of the models (Table 1), with the exception of $\alpha_{PT}$, which was typically lower. Other studies also found $\alpha_{PT}$ values far below 1.26 but within the range of our study, mainly for forests (e.g. Komatsu, 2005;Shuttleworth and Calder, 1979;Viswanadham et al., 1991;Eaton et al., 2001) but also for tundra (Eaton et

al., 2001) or grassland sites (Katerji and Rana, 2011) – see McMahon et al. (2013) for an overview. Our results and these studies demonstrate that the standard level of $\alpha_{PT}$=1.26 is close to the upper threshold of evaporation and will overestimate $E_p$ at most sites (Table 5).

**4.3 Performance of the Penman-Monteith method**

The poor performance of the $PM_r$, $PM_s$ and $PM_b$ methods was relatively unexpected. Because the Penman-

Monteith method incorporates the effects of air temperature, humidity, radiation and wind, it is often considered superior (e.g. Sheffield et al., 2012), and is even used as reference method to evaluate other methods (e.g. Sentelhas et al., 2010;Xu and Singh, 2002;Oudin et al., 2005b). However, in studies estimating $E_a$, in which $g_c$ is adjusted so it reflects the actual rather than potential situation, the Penman-Monteith method has already been shown to perform worse than other, simpler methods, such as the Priestley and Taylor method (e.g. Michel et al.,

2016;Ershadi et al., 2014). Its performance depends on the reliability of the wide range input data required, and on the methods used to derive $r_{aH}$ and $g_c$ (Seiller and Anctil, 2016;Singh and Xu, 1997;Dolman et al., 2014). In our case, the strict procedure followed to select the samples of 107 eddy-covariance datasets (see Sect. 2.2) ensured that all relevant variables were available, and that these meteorological measurements could be considered of high quality. Hence, in our analysis, poor input quality is unlikely to be the cause of low performance.

We believe that the underlying assumption of a constant $g_{c\_ref}$ under no stress typically adopted by PM methods ($PM_r$, $PM_s$, $PM_b$) when estimating $E_p$, is a more likely explanation of the poor performance. PM was the only method of which the biome-specific calibration did not improve its performance. This is partially because of the large variation in $g_{c\_ref}$-values between the different flux sites of the same biome type (Table 2). In addition, of all the key parameters, $g_{c\_ref}$ values had the largest mean relative standard deviation of the unstressed datasets of the

individual sites (*results not shown*). Canopy conductance of the unstressed dataset was significantly negatively correlated with VPD in 45% of the sites (Fig. 3b, Table 3). The relationship between $g_c$ and VPD for two such sites is illustrated in Fig. 5. It is clear that the $g_c$ data of unstressed days (red dots) are among the highest $g_c$ values for a given VPD, illustrating the validity of the energy balance method. However, it is also clear that $g_c$ of these



unstressed days decreases sharply with increasing VPD. As a consequence, their mean value, used to later calculate $g_{c\_ref}$ per biome type, is highly influenced by the local VPD and is not necessarily a representative value for this ecosystem.

The dependence of $g_c$ on VPD (e.g. Sumner and Jacobs, 2005;Granier et al., 2000;Jones, 1992;Novick et al., 2016)
has been well studied and incorporated in most conventional stomatal or canopy conductance models (e.g. Leuning, 1995;Jarvis, 1976;Ball et al., 1987). Yet, out of practical reasons, $g_{c\_ref}$ is usually taken as a constant in $E_p$-methods using the Penman-Monteith approach, with the $PM_r$ as best illustration. Our data confirm that stomata close when VPD increases, even in unstressed conditions. As such, the VPD-dependence of $g_c$ smoothens the impact of VPD in the Penman-Monteith equation: if VPD gets high, $g_c$ becomes very low, limiting the impact of
VPD on $E_a$ (Eq. 1). Assuming a constant $g_c$ value overestimates the influence of VPD (and wind speed) on $E_p$. Schymanski and Or (2017) recently pointed at a possible error in the Penman-Monteith formulation at leaf scale and claimed that a similar issue is present at the ecosystem scale application of the Penman-Monteith method. In that case, a second key parameter needs to be added, relating to the whether leaves are amphi-, epi- or hypostomatous. As Eq. 1 is the standard equation used in all $E_p$ methods, we choose not to incorporate this
correction (or, we assumed that the canopy can be represented as a big leaf with no latent and heat flux exchange at the lower part, a standard assumption of the Penman-Monteith equation). However, even if this correction would be incorporated, the assumption of a constant $g_c$ value would lead to similar issues ad would likely make the PM-methods less performant.

Apart from the VPD-dependence, taking a constant $g_c$-value in the Penman-Monteith method also ignores the
effect of increasing $CO_2$ levels on $g_c$. As a result, Milly and Dunne (2016) found that the Penman-Monteith methods with constant $g_c$ overpredicted $E_p$ in models predicting future water use. Incorporating a VPD and $CO_2$ calibration of $g_c$ in the Penman-Monteith equation is outside the scope of this study, but can be an interesting approach to further improve $E_p$ calculations. However, it would make the model even more complex and require more empirical input data. Likewise, taking a wet canopy as reference in the Penman method ($g_c = \infty$ or $r_c=0$), not
only severely overestimates $E_p$ (Table 6) but also overestimates the influence of VPD and wind speed on $E_p$.

*(Insert Figure 5)*

### 4.4 Possible issues with $PT_b$ and $MD_b$ methods

The simpler Priestley and Taylor and radiation-based methods came forward as the best methods for assessing $E_p$ with both criteria to define unstressed days. Both $PT_b$ and $MD_b$ are attractive from a modelling perspective, as they require a minimum of input data. However, this simplicity can also hold risks. The Priestley and Taylor method has been criticised for the implicit assumptions, which are also present in the $MD_b$-method. For instance, by not incorporating wind speed explicitly, it is assumed that the effect of wind speed on $E_p$ is somehow embedded
within $\alpha_{PT}$ (or $\alpha_{MD}$).Yet, several studies indicate that wind speeds are decreasing ('stilling') globally (McVicar et al., 2008;McVicar et al., 2012;Vautard et al., 2010). McVicar et al. (2012) also reported an associated decreasing trend in observed pan evaporation worldwide as well as in $E_p$ calculated with the $PM_r$ method. As Priestley and Taylor methods do not incorporate effects of wind speed McVicar et al. (2012) argued that these should not be used in climate models.



A separate question is whether more complex $E_p$ methods that incorporate the effects of wind speed or VPD do this correctly; the above-mentioned issues about the fixed parameterisation of the Penman-Monteith methods for estimating $E_p$ indicate that this is typically not the case. In addition, both Penman methods showed a relatively poor performance and severely overestimated $E_{unstr}$ (Tables 4 and 6; S2 and S4).

Regarding the non-consideration of wind speed by simpler methods, our records shows a limited effect of $u$ on $E_a$ and $E_p$, even when considering larger temporal scales. Of the 16 flux towers with at least 10 years of evaporation data, we calculated the yearly average $E_a$ as well as the annual mean climatic variables. Yearly averages were calculated from monthly averages, which on their turn were calculated if at least three daytime measurements were available. Despite a relatively large mean standard deviation in yearly average $u$ of 7.0%, yearly average $u$

was not significantly correlated with the $E_a$ in any of these sites. In contrast, yearly average net radiation was positively correlated with evaporation for seven of the 16 sites, with comparable mean standard deviation in annual $R_n$ (8.5%). Moreover, looking at a daily scale and at $E_p$, neither $\alpha_{MD}$ nor $\alpha_{PT}$ were heavily influenced by wind speed (Fig. 3c, d, Table 3). Hence, the implicit assumption of the independence of $\alpha_{PT}$ and $\alpha_{MD}$ on wind speed temporal variability seems legitimate. In fact, given the fact that $\alpha_{MD}$ was hardly affected by any climatic variable,

and given the relatively small range in $\alpha_{MD}$ values within each biome (Table 2), it appears that $\alpha_{MD}$ is a robust biome property that can be used in the upscaling of these methods for their global application. The robustness of $\alpha_{MD}$ as biome property is furthermore confirmed by the analysis with independent calibration and validation sites, which hardly affected the unRMSE and bias (Tables S9-10). All the well performing methods, and particularly the two best methods (MD$_b$ and PT$_b$), rely heavily on (R$_n$-G) (Eqs. 2 and 3). In Section 3.2, all $E_p$ calculations

used (R$_n$-G) obtained during unstressed conditions. The question is whether $E_p$ can also be calculated correctly using the observed (R$_n$-G) when the ecosystems are not unstressed. As mentioned in Section 1, while several authors argued that the net radiation of the reference system should be the one measured for the actual ecosystem (e.g. Rind et al., 1990;Crago and Crowley, 2005;Granger, 1989), others considered only incoming shortwave and longwave radiation as forcing variables and outgoing longwave and shortwave radiation as ecosystem responses

(e.g. Lhomme, 1997;Lhomme and Guilioni, 2006). Despite practical implications, it is clear that T$_s$ - hence, outgoing longwave radiation - is lower when ecosystems have higher evaporation and are less stressed (Maes and Steppe, 2012). Therefore, it appears appropriate not to use the shortwave or longwave outgoing radiation, or G, to compute $E_p$, but to use values of $\alpha$, T$_s$ and G that reflect unstressed conditions for estimating (R$_n$-G). A method to derive these variables based on flux tower data of the unstressed datasets is presented in the Section S2 of the

supporting information. However, as this method requires a large amount of input data, it is not practically applicable at global scale. Comparing $E_p$ obtained with this correct method with $E_p$ calculated with the actual (R$_n$-G) reveal that the latter method underestimates $E_p$ with 8.2 ± 10.1%. There are no significant differences between biomes, but the distribution of the underestimation is left-skewed: although the median underestimation is 5.5%, the underestimation was larger than 10% in 22% of the sites (Fig. 6).

The main reason for this underestimations is the difference between T$_s$ of the actual and of the unstressed system, which accounted for 65% of the underestimation. If the unstressed T$_s$ is estimated as the mean of T$_a$ and the actual T$_s$, $E_p$ can be calculated with the MD$_b$ method as

$$E_p = \alpha_{MD} \left( (1 - \alpha)\, SW_{in} + \varepsilon\, LW_{in} - 0.5\, \varepsilon\, LW_{out} - 0.5\, \varepsilon\, \sigma\, T_a^4 - G \right) \tag{16}$$





This approach results in a slight mean underestimation of $E_p$ of 2.6 ± 5.8%, but with a mean median value at -0.1% (Fig. 6). Given the low error and the straightforward calculation, we recommend this method to calculate $E_p$ at global scales.

5    *(Insert Figure 6)*

**Conclusion**

Based on a large sample of eddy-covariance sites, we demonstrated that simple energy-driven methods calibrated per biome type can estimate $E_p$ with a higher accuracy than more complex Penman-Monteith approaches. This was consistent across all 11 biomes represented in the database, and for two different criteria to identify unstressed

10    days, based on soil moisture and evaporative fraction thresholds. Our analyses also showed that using the net radiation corrected for unstressed soil moisture conditions, allows to palliate an underestimation of $E_p$ in energy-driven methods, and confirmed that the key parameters required to apply the higher-performance energy-driven methods are relatively insensitive to climate forcing or $CO_2$ levels. This makes these methods robust for incorporation into global hydrological, drought or climate models. Finally, we conclude that Penman-Monteith

methods for estimating $E_p$ should only be prioritised if the maximum stomata conductance is calculated dynamically.

**Data availability**

The FLUXNET2015 dataset can be downloaded from http://fluxnet.fluxdata.org/data/fluxnet2015-dataset/. We provided the main script for calculating potential evaporation with the different method as well as the daily flux

data of one site (AU-How), for which permission of distribution was granted. For further questions, we ask researchers to contact the corresponding author (wh.maes@ugent.be).

**Acknowledgements**

This study was funded by the Belgian Science Policy Office (BELSPO) in the frame of the STEREO III program project STR3S (SR/02/329). Diego G. Miralles acknowledges support from the European Research Council

(ERC) under grant agreement no. 715254 (DRY–2–DRY). This work used eddy covariance data acquired and shared by the FLUXNET community, including these networks: AmeriFlux, AfriFlux, AsiaFlux, CarboAfrica, CarboEuropeIP, CarboItaly, CarboMont, ChinaFlux, FLUXNET-Canada, GreenGrass, ICOS, KoFlux, LBA, NECC, OzFlux-TERN, TCOS-Siberia, and USCCC. The ERA-Interim reanalysis data are provided by ECMWF and processed by LSCE. The FLUXNET eddy covariance data processing and harmonization was carried out by

the European Fluxes Database Cluster, AmeriFlux Management Project, and Fluxdata project of FLUXNET, with the support of CDIAC and ICOS Ecosystem Thematic Center, and the OzFlux, ChinaFlux and AsiaFlux offices. The Atqasuk and Ivotuk towers in Alaska were supported by the Office of Polar Programs of the National Science Foundation (NSF) awarded to DZ, WCO, and DAL (award number 1204263) with additional logistical support funded by the NSF Office of Polar Programs, and by the Carbon in Arctic Reservoirs Vulnerability Experiment

(CARVE), an Earth Ventures (EV-1) investigation, under contract with the National Aeronautics and Space





Administration, and by the ABoVE (NNX15AT74A; NNX16AF94A) Program. The OzFlux network is supported by the Australian Terrestrial Ecosystem Research Network (TERN, http://www.tern.org.au).

**Author contribution.** WHM and DGM designed the research; PG and NECV assisted in developing the optimal method for analysing all flux tower data; WHM performed the calculations and analyses and prepared the manuscript, with contributions from all co-authors.



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

45



**Table 1. Overview of the different $E_p$ methods used in this study and their calculation.**

| | | Key parameter | $R_n$ | | $r_{aH}$ | $T_a$ | RH or VPD | N |
|---|---|---|---|---|---|---|---|---|
| | | | SW* | LW* | | | | |
| **Penman-Monteith** | | $g_{C\_ref}$ (mm s⁻¹) | | | | | | |
| $PM_r$ | Reference crop | 14.49 | FAO-56 ($\alpha$=0.23) | FAO-56 $f(T_{max},T_{min}, SW, SWTOA, e_a)$ | 208 $u_{2m}^{-1}$ | From $T_{max},T_{min}$ | From $RH_{Max}, RH_{Min}$ | FAO-56 |
| $PM_s$ | Standard | 14.49 | measured | measured | calculated | Daytime mean | Daytime mean | |
| $PM_b$ | Biome-specific | Biome-specific | measured | measured | calculated | Daytime mean | Daytime mean | |
| **Penman** | | $g_{C\_ref}$ (mm s⁻¹) | | | | | | |
| $Pe_r$ | Reference crop | $\infty$ ($r_{c\_ref}$ = 0) | FAO-56 ($\alpha$=0.23) | FAO-56 $f(T_{max},T_{min}, SW, SWTOA, e_a)$ | 208 $u_{2m}^{-1}$ | From $T_{max},T_{min}$ | From $RH_{Max}, RH_{Min}$ | |
| $Pe_s$ | Standard | $\infty$ ($r_{c\_ref}$ = 0) | measured | measured | calculated | Daytime mean | Daytime mean | |
| **Priestley and Taylor** | | $\alpha_{PT}$ (-) | | | | | | |
| $PT_r$ | Reference crop | 1.26 | FAO-56 ($\alpha$=0.23) | FAO-56 $f(T_{max},T_{min}, SW, SWTOA, e_a)$ | | | | |
| $PT_s$ | Standard | 1.26 | measured | measured | calculated | Daytime mean | Daytime mean | |
| $PT_B$ | Biome-specific | Biome-specific | measured | measured | calculated | Daytime mean | Daytime mean | |
| **Milly and Dunne** | | $\alpha_{RB}$ (-) | | | | | | |
| $MD_r$ | Reference crop | 0.8 | FAO-56 ($\alpha$=0.23) | FAO-56 $f(T_{max},T_{min}, SW, SWTOA, e_a)$ | | | | |
| $MD_s$ | Standard | 0.8 | measured | measured | calculated | Daytime mean | Daytime mean | |
| $MD_B$ | Biome-specific | Biome-specific | measured | measured | calculated | Daytime mean | Daytime mean | |
| **Thorntwaite** | | $\alpha_{Th}$ (-) | | | | | | |
| $Th_s$ | Standard | 16 | | | | From $T_{max},T_{min}$ | | Measured |
| $Th_b$ | Biome-specific | Biome-specific | | | | From $T_{max},T_{min}$ | | Measured |





| Oudin | | $\alpha_{ou}$ | | |
|---|---|---|---|---|
| Ou$_s$ | Standard | 100 | | Daily mean |
| Ou$_b$ | Biome-specific | Biome-specific | | Daily mean |

N=number of daylight hours; $T_{max}$, $T_{min}$, $RH_{max}$ and $RH_{min}$ the maximum and minimum daily air temperature or RH; SW* and LW* are net shortwave and net longwave radiation; SWTOA is the shortwave incoming radiation at the top the atmosphere; FAO-56 refers to the methodology described by Allen et al. (1998)





**Table 2. Overview of the difference of the key parameters ($g_{c\_ref}$, $\alpha_{PT}$, $\alpha_{MD}$, $\alpha_{Th}$ and $\alpha_{Ou}$) during unstressed conditions per biome. The energy balance method was used for defining unstressed days (See section 2.4, see Table 1 for definition of key parameters). The $p$ value of the ANOVA test is given, as well as the mean ± standard deviation for each biome. Letters in subscript indicate statistically significant differences (Tukey post hoc test, $p<0.05$). The number of sites per biome is given between brackets.**

| | $g_{c\_ref}$ (mm s$^{-1}$) | $\alpha_{PT}$ (-) | $\alpha_{MD}$ (-) | $\alpha_{Th}$ (-) | $\alpha_{Ou}$ (-) |
|---|---|---|---|---|---|
| $p$ (ANOVA) | 0.017 | 0.004 | <0.001 | <0.001 | <0.001 |
| CRO (10) | 38.3 ± 23.0 | 1.15 ± 0.14[a] | 0.86 ± 0.09[a] | 38.7 ± 14.5[ab] | 77.0 ± 27.8[b] |
| GRA (20) | 30.5 ± 40.2 | 1.02 ± 0.16[ab] | 0.74 ± 0.12[ab] | 30.4 ± 13.9[ab] | 103.2 ± 38.9[ab] |
| DBF (15) | 32.6 ± 27.4 | 1.09 ± 0.14[ab] | 0.80 ± 0.08[ab] | 33.3 ± 7.8[ab] | 70.5 ± 18.0[ab] |
| EBF (9) | 42.0 ± 36.6 | 1.09 ± 0.15[ab] | 0.74 ± 0.05[a] | 53.1 ± 16.8[a] | 95.5 ± 22.9[ab] |
| ENF (26) | 28.4 ± 52.1 | 0.89 ± 0.26[ab] | 0.62 ± 0.09[ab] | 40.3 ± 16.7[ab] | 92.0 ± 21.8[ab] |
| MF (4) | 10.0 ± 7.1 | 0.88 ± 0.23[b] | 0.64 ± 0.13[b] | 26.1 ± 3.6[ab] | 138.2 ± 91.6[ab] |
| CSH (2) | 8.5 ± 3.9 | 0.90 ± 0.10[ab] | 0.64 ± 0.15[ab] | 41.4 ± 13.7[ab] | 130.3 ± 36.1[a] |
| WSA (5) | 8.4 ± 3.4 | 0.95 ± 0.09[ab] | 0.70 ± 0.10[ab] | 33.8 ± 6.4[ab] | 104.6 ± 19.7[ab] |
| SAV (6) | 7.8 ± 3.7 | 0.87 ± 0.14[ab] | 0.68 ± 0.15[b] | 35.0 ± 4.1[ab] | 147.1 ± 63.9[ab] |
| OSH (5) | 4.3 ± 2.0 | 0.79 ± 0.11[b] | 0.58 ± 0.09[ab] | 31.3 ± 11.2[ab] | 147.7 ± 61.8[ab] |
| WET (5) | 20.0 ± 14.1 | 1.03 ± 0.47[ab] | 0.75 ± 0.11[ab] | 17.8 ± 13.3[b] | 638.6 ± 1230.1[ab] |

CRO=cropland; DBF=Deciduous Broadleaf Forest; EBF=Evergreen Broadleaf Forest; ENF=Evergreen Needle Forest; MF=Mixed Forest; CSH=Closed Shrubland; WSA=Woody Savanna; SAV=Savanna; OSH=Open Shrubland; GRA=Grasslands; WET=Wetlands





**Table 3. Influence of climate forcing variables on $E_{unstr}$ and selected key parameters ($g_{c\_ref}$, $\alpha_{PT}$, $\alpha_{MD}$). (left) Mean ± 1 standard deviation of the correlations of $E_{unstr}$, $g_c$, $\alpha_{PT}$ and $\alpha_{MD}$ against the climate forcing variables, and (right) number of sites (out of total = 107) with significant negative/positive correlations between $E_{unstr}$, $\alpha_{PT}$, $g_{c\_ref}$ and $\alpha_{MD}$ and the climate forcing variables. Based on unstressed days only defined using the energy balance criterion.**

| | Mean ± 1 standard deviation of the correlations | | | | Number of sites with significant negative/positive correlations | | | |
|---|---|---|---|---|---|---|---|---|
| | $E_{unstr}$ | $g_{c\_ref}$ | $\alpha_{PT}$ | $\alpha_{MD}$ | $E_{unstr}$ | $g_{c\_ref}$ | $\alpha_{PT}$ | $\alpha_{MD}$ |
| Wind | $0.13 \pm 0.26$ | $0.03 \pm 0.25$ | $0.12 \pm 0.31$ | $0.01 \pm 0.22$ | 6/26 | 4/13 | 11/30 | 5/6 |
| $T_{air}$ | $0.60 \pm 0.24^*$ | $-0.22 \pm 0.29$ | $-0.21 \pm 0.34$ | $-0.02 \pm 0.28$ | 0/93 | 43/0 | 43/5 | 16/13 |
| VPD | $0.64 \pm 0.20^*$ | $-0.27 \pm 0.27$ | $-0.11 \pm 0.31$ | $-0.01 \pm 0.28$ | 0/93 | 48/0 | 31/10 | 15/11 |
| $R_n$ | $0.90 \pm 0.08^*$ | $-0.05 \pm 0.25$ | $-0.13 \pm 0.30$ | $-0.10 \pm 0.31$ | 0/106 | 17/3 | 33/5 | 30/14 |
| $[CO_2]$ | $-0.16 \pm 0.30$ | $-0.01 \pm 0.23$ | $-0.03 \pm 0.22$ | $-0.03 \pm 0.25$ | 34/5 | 7/5 | 9/4 | 12/4 |

$^*$significantly different from 0





**Table 4. Mean correlations per biome between the measured $E_{unstr}$ and the different $E_p$ methods. The methods with the highest correlation per biome are highlighted in bold and underlined. Based on unstressed days only defined using the energy balance criterion.**

| | Radiation, Temperature, VPD | | | | | Radiation, Temperature | | | Radiation | | | Temperature | | | |
| | Penman-Monteith | | | Penman | | Priestley and Taylor | | | Milly and Dunne | | | Thornthwaite | | Oudin | |
| | Ref. crop | Standard | Biome | Ref. crop | Standard | Ref. crop | Standard | Biome | Ref. crop | Standard | Biome | Standard | Biome | Standard | Biome |
|---|---|---|---|---|---|---|---|---|---|---|---|---|---|---|---|
| CRO (10) | 0.84 | 0.91 | 0.90 | 0.76 | 0.81 | 0.86 | 0.96 | 0.96 | 0.82 | **_0.96_** | **_0.96_** | 0.77 | 0.77 | 0.74 | 0.74 |
| GRA (20) | 0.79 | 0.87 | 0.87 | 0.77 | 0.84 | 0.82 | 0.93 | 0.93 | 0.80 | **_0.94_** | **_0.94_** | 0.55 | 0.54 | 0.55 | 0.55 |
| DBF (15) | 0.78 | 0.87 | 0.88 | 0.79 | 0.85 | 0.78 | 0.91 | 0.91 | 0.75 | **_0.91_** | **_0.91_** | 0.57 | 0.56 | 0.57 | 0.57 |
| EBF (9) | 0.88 | 0.89 | 0.88 | 0.86 | 0.85 | 0.87 | **_0.91_** | **_0.91_** | 0.83 | 0.90 | 0.90 | 0.71 | 0.79 | 0.57 | 0.57 |
| ENF (26) | 0.89 | 0.90 | 0.91 | 0.88 | 0.86 | 0.90 | 0.95 | 0.95 | 0.88 | **_0.95_** | **_0.95_** | 0.77 | 0.79 | 0.76 | 0.76 |
| MF (4) | 0.90 | 0.93 | 0.93 | 0.90 | 0.93 | 0.90 | **_0.94_** | **_0.94_** | 0.88 | 0.93 | 0.93 | 0.79 | 0.75 | 0.74 | 0.74 |
| CSH (2) | 0.90 | 0.94 | 0.93 | 0.89 | 0.90 | 0.90 | 0.95 | 0.95 | 0.89 | **_0.95_** | **_0.95_** | 0.80 | 0.78 | 0.75 | 0.75 |
| WSA (5) | 0.76 | 0.78 | 0.78 | 0.73 | 0.73 | 0.80 | 0.89 | 0.89 | 0.79 | **_0.90_** | **_0.90_** | 0.41 | 0.41 | 0.46 | 0.46 |
| SAV (6) | 0.79 | 0.82 | 0.81 | 0.77 | 0.79 | 0.83 | **_0.91_** | **_0.91_** | 0.81 | 0.91 | 0.91 | 0.52 | 0.52 | 0.56 | 0.56 |
| OSH (5) | 0.72 | 0.80 | 0.78 | 0.64 | 0.78 | 0.79 | 0.90 | 0.90 | 0.77 | **_0.90_** | **_0.90_** | 0.54 | 0.53 | 0.56 | 0.56 |
| WET (5) | 0.87 | 0.81 | 0.76 | 0.87 | 0.66 | 0.79 | 0.83 | 0.83 | 0.68 | **_0.85_** | **_0.85_** | 0.50 | 0.45 | 0.61 | 0.61 |
| Overall (107) | 0.83 | 0.87 | 0.87 | 0.81 | 0.83 | 0.84 | 0.92 | 0.92 | 0.81 | **_0.93_** | **_0.93_** | 0.62 | 0.63 | 0.63 | 0.63 |





**Table 5. Unbiased Root Mean Square Error (UnRMSE) (in mm day$^{-1}$) for the $E_p$ methods per biome. The methods with the lowest UnRMSE per biome are indicated in bold and are underlined. Based on unstressed days only defined using the energy balance criterion.**

| | *Radiation, Temperature, VPD* | | | | | *Radiation, Temperature* | | | *Radiation* | | | *Temperature* | | | |
| | Penman-Monteith | | | Penman | | Priestley and Taylor | | | Milly and Dunne | | | Thornthwaite | | Oudin | |
| | Ref. crop | Standard | Biome | Ref. crop | Standard | Ref. crop | Standard | Biome | Ref. crop | Standard | Biome | Standard | Biome | Standard | Biome |
|---|---|---|---|---|---|---|---|---|---|---|---|---|---|---|---|
| CRO (10) | 1.16 | 0.79 | 1.04 | 1.60 | 2.88 | 1.27 | 0.62 | 0.58 | 1.21 | 0.57 | **0.55** | 1.24 | 1.24 | 1.29 | 1.27 |
| GRA (20) | 1.22 | 0.70 | 0.81 | 1.75 | 1.04 | 1.40 | 0.58 | 0.47 | 1.13 | 0.44 | **0.44** | 1.07 | 1.03 | 1.05 | 1.04 |
| DBF (15) | 1.14 | 0.88 | 0.89 | 1.21 | 1.36 | 1.29 | 0.75 | **0.72** | 1.20 | 0.72 | 0.72 | 1.41 | 1.42 | 1.37 | 1.32 |
| EBF (9) | 0.84 | 0.62 | 0.93 | 1.07 | 1.33 | 1.09 | 0.75 | 0.59 | 0.96 | 0.55 | **0.54** | 1.04 | 0.98 | 1.15 | 1.14 |
| ENF (26) | 0.98 | 0.78 | 0.99 | 1.20 | 14.89 | 1.26 | 0.84 | 0.52 | 1.09 | 0.59 | **0.50** | 0.94 | 0.91 | 0.96 | 0.97 |
| MF (4) | 1.23 | 0.69 | 0.69 | 1.58 | 1.11 | 1.64 | 0.86 | **0.58** | 1.26 | 0.64 | 0.59 | 1.11 | 1.03 | 1.03 | 0.99 |
| CSH (2) | 0.82 | 0.59 | 0.59 | 0.98 | 0.92 | 1.12 | 0.75 | **0.48** | 0.91 | 0.55 | 0.49 | 0.90 | 0.96 | 0.83 | 0.81 |
| WSA (5) | 1.15 | 0.93 | 0.80 | 1.41 | 1.68 | 1.27 | 0.67 | 0.52 | 1.00 | 0.53 | **0.51** | 1.10 | 1.10 | 0.99 | 0.99 |
| SAV (6) | 1.22 | 1.02 | 0.83 | 1.53 | 1.88 | 1.39 | 0.76 | 0.52 | 1.07 | 0.58 | **0.52** | 1.22 | 1.21 | 1.10 | 0.97 |
| OSH (5) | 1.37 | 0.73 | 0.63 | 1.94 | 0.92 | 1.63 | 0.67 | **0.43** | 1.28 | 0.48 | 0.44 | 1.12 | 1.03 | 0.90 | 0.80 |
| WET (5) | 1.27 | 1.25 | 1.38 | 1.38 | 4.14 | 1.72 | 1.28 | 1.13 | 1.91 | 1.14 | **1.10** | 2.20 | 2.29 | 1.65 | 2.01 |
| Overall (107) | 1.11 | 0.80 | 0.91 | 1.41 | 4.86 | 1.34 | 0.75 | 0.57 | 1.16 | 0.60 | **0.56** | 1.16 | 1.14 | 1.12 | 1.11 |





**Table 6. Mean bias (in mm day$^{-1}$) for the $E_p$ methods per biome. The best performing method per biome is indicated in bold and is underlined. Based on unstressed days only defined using the energy balance criterion.**

| | *Radiation, Temperature, VPD* | | | | | *Radiation, Temperature* | | | *Radiation* | | | *Temperature* | | | |
| | Penman-Monteith | | | Penman | | Priestley and Taylor | | | Milly and Dunne | | | Thornthwaite | | Oudin | |
| | Ref. crop | Standard | Biome | Ref. crop | Standard | Ref. crop | Standard | Biome | Ref. crop | Standard | Biome | Standard | Biome | Standard | Biome |
| CRO (10) | 1.14 | -0.49 | 0.84 | 2.83 | 2.64 | 2.20 | 0.47 | **-0.01** | 1.43 | -0.24 | 0.12 | -0.65 | -0.59 | -1.62 | -0.62 |
| GRA (20) | 2.65 | 0.53 | 1.16 | 4.37 | 1.90 | 3.69 | 1.11 | **0.02** | 2.57 | 0.22 | -0.10 | -0.14 | -0.44 | -0.61 | -0.73 |
| DBF (15) | 0.30 | -0.48 | 0.89 | 1.30 | 2.63 | 1.81 | 0.94 | **-0.06** | 0.74 | -0.13 | -0.15 | -1.94 | -2.03 | -2.44 | -0.71 |
| EBF (9) | 0.70 | **0.04** | 0.95 | 1.39 | 1.74 | 1.39 | 0.79 | 0.16 | 0.79 | 0.17 | -0.13 | -0.83 | -0.27 | -0.53 | -0.36 |
| ENF (26) | 1.28 | 0.45 | 1.23 | 2.03 | 1.04 | 2.06 | 1.17 | -0.05 | 1.88 | 0.90 | **0.02** | -0.15 | -0.05 | -0.73 | -0.54 |
| MF (4) | 2.22 | 0.65 | 0.30 | 3.31 | 2.04 | 3.26 | 1.46 | -0.07 | 2.53 | 0.87 | -0.04 | 0.73 | 0.19 | **-0.01** | -0.99 |
| CSH (2) | 1.01 | 0.49 | **0.00** | 1.61 | 1.79 | 1.46 | 1.10 | -0.04 | 0.92 | 0.51 | -0.14 | 0.18 | 0.39 | 0.14 | -0.56 |
| WSA (5) | 2.67 | 1.16 | 0.17 | 3.68 | 3.88 | 3.63 | 1.42 | **-0.03** | 2.33 | 0.40 | -0.22 | -0.14 | -0.23 | -0.20 | -0.39 |
| SAV (6) | 2.56 | 1.30 | 0.31 | 3.57 | 3.78 | 3.34 | 1.47 | -0.15 | 2.21 | 0.54 | -0.13 | 0.03 | **0.00** | 0.40 | -0.93 |
| OSH (5) | 4.32 | 1.68 | 0.37 | 6.20 | 2.73 | 5.08 | 2.00 | 0.10 | 3.89 | 1.15 | **0.00** | 1.13 | 0.84 | 0.81 | -0.44 |
| WET (5) | 2.34 | 1.28 | 1.74 | 4.17 | 4.51 | 3.45 | 2.00 | 1.04 | 3.29 | 1.43 | 1.12 | 1.42 | 0.29 | **-0.52** | -2.79 |
| Overall (107) | 1.69 | 0.40 | 0.93 | 2.88 | 2.21 | 2.67 | 1.14 | 0.04 | 1.92 | 0.45 | **0.00** | -0.38 | -0.45 | -0.80 | -0.72 |





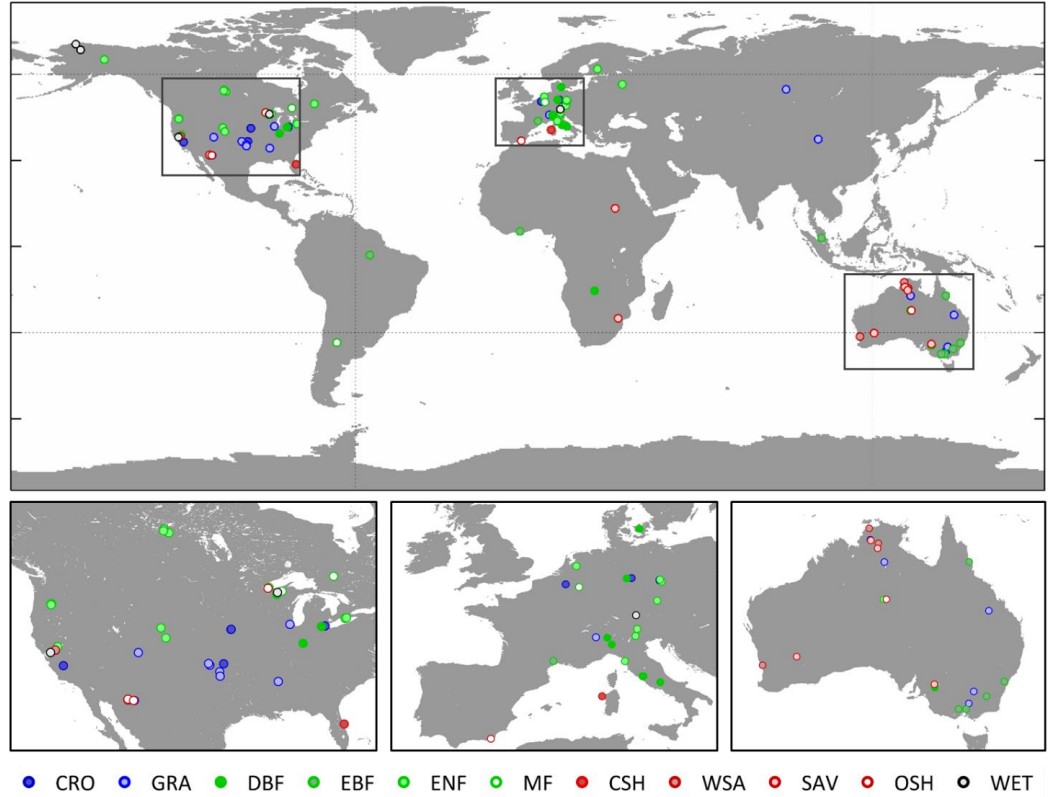

**Figure 1. Location of the flux sites used in this study per biome. CRO=cropland; DBF=Deciduous Broadleaf Forest; EBF=Evergreen Broadleaf Forest; ENF=Evergreen Needle Forest; MF=Mixed Forest; CSH=Closed Shrubland; WSA=Woody Savannah; SAV=Savannah; OSH=Open Shrubland; GRA=Grasslands; WET=Wetlands.**



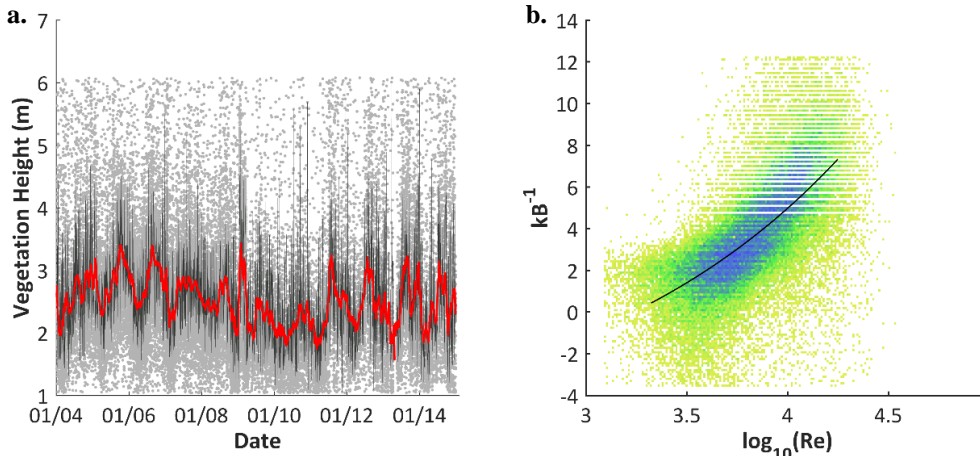

**Figure 2. (a) Vegetation height dynamics in time (grey dots: half-hourly measurements; dark grey lines: daily mean vegetation height; red line: 30-day moving average (i.e. the final vegetation height dataset). (b) Relation between the Stanton number ($k$B$^{-1}$) and the Reynolds number (Re). Both plots correspond to the woody savannah site of Santa Rita Mesquite (Arizona, USA).**





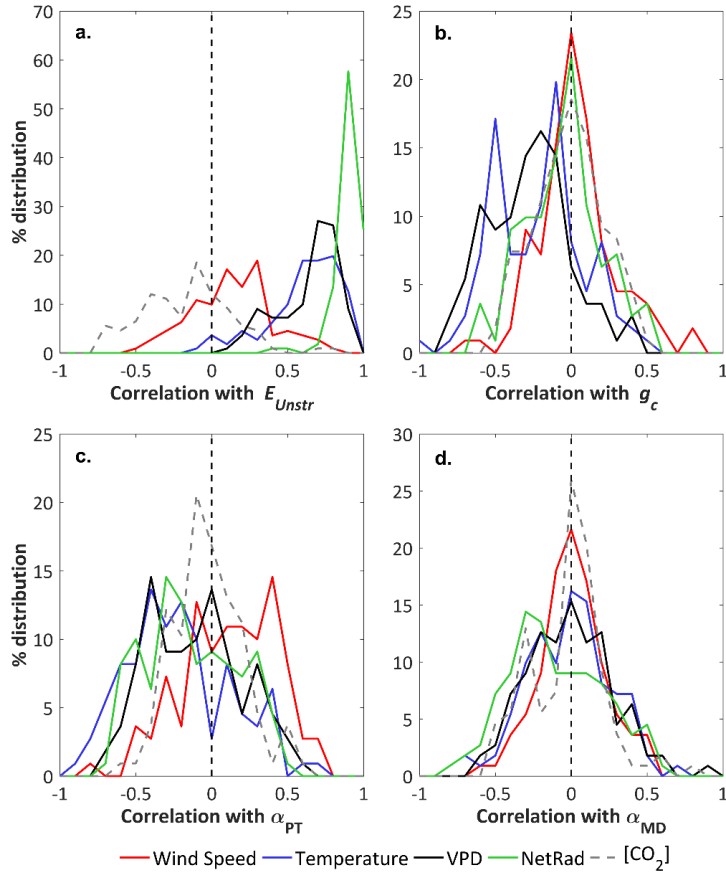

**Figure 3. Histograms of correlations between the climate forcing variables and selected key parameters (a)** $E_{unstr}$ **(b)** $g_{c\_ref}$**, (c)** $\alpha_{PT}$ **and (d)** $\alpha_{MD}$ **measured in all flux tower sites. Based on unstressed days only defined using the energy balance criterion.**

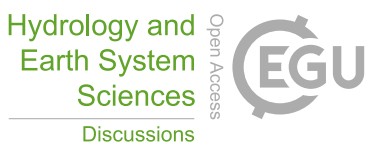



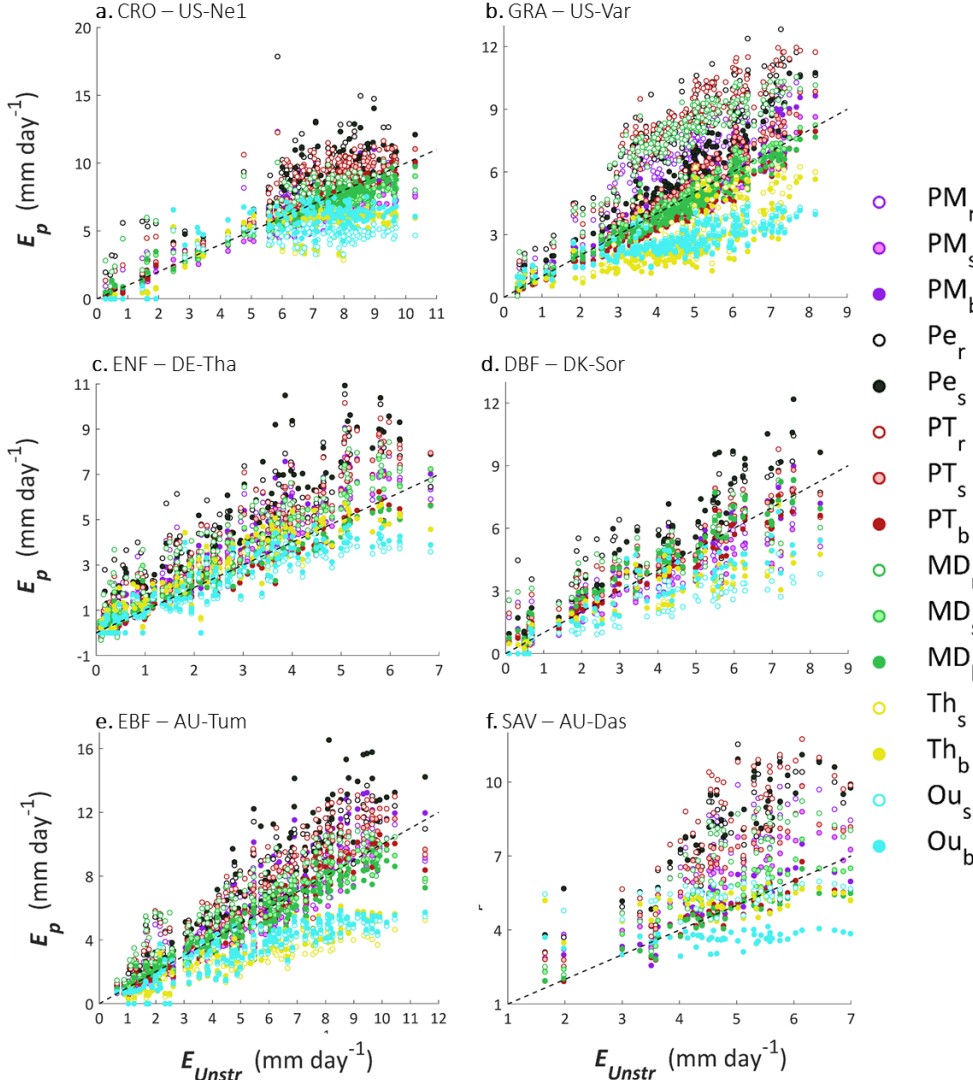

**Figure 4. Scatterplot of the measured $E_{unstr}$ versus $E_p$ calculated with the different methods. The discontinuous line is the 1:1 line. Based on unstressed days only defined using the energy balance criterion.**





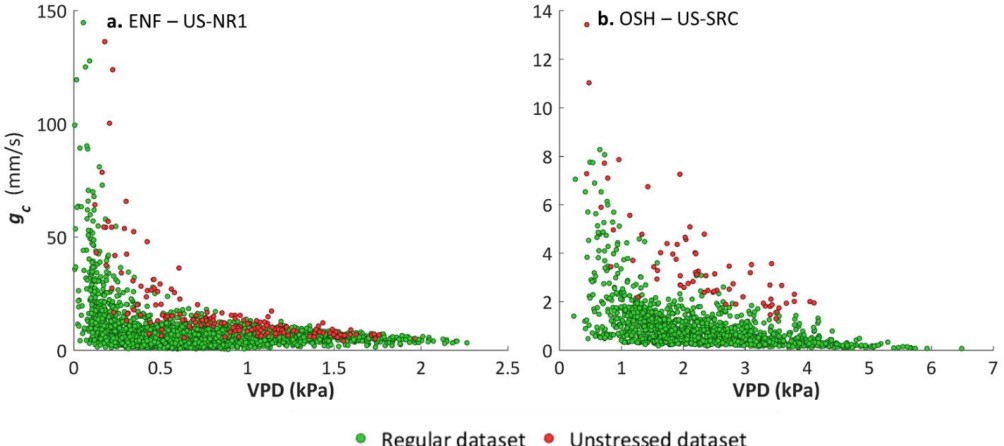

**Figure 5. Canopy conductance $g_c$ as a function of vapour pressure deficit (VPD) of the regular and the unstressed datasets of two flux sites, (a) the evergreen needle forest Niwot Ridge Forest and (b) the open savannah woodland site Santa Rita Creosote.**



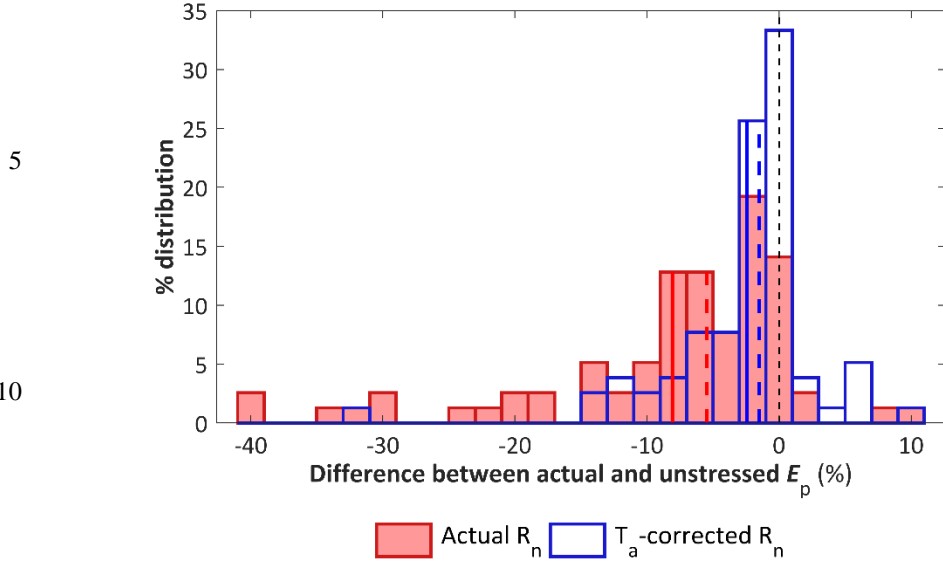

**Figure 6. The error in the estimate of $E_p$ of two empirical methods to calculate unstressed ($R_n$-G). The first empirical method simply take the actual ($R_n$-G) as input, the second method corrects the actual ($R_n$-G) with $T_a$ (Eq. (16)). Negative values indicate an underestimation of the empirical methods. Full vertical lines indicate the mean and dotted vertical lines the median values of each method.**

