# Peer review of "Potential evaporation at eddy-covariance sites across the globe"

_Hydrology and Earth System Sciences, 2017_

## Short Comment (SC1) · 21 Feb 2018

Hey guys, we actually did this study back in 2012, presented at AGU in a session chaired by Martha Anderson, but we never ended up publishing it. No worries though– we snooze we lose! But, it would be nice if you wouldn't mind citing that presentation? It was led by a high-school student, so I think she'd be thrilled to be cited.

Palmer, C., Fisher, J.B., Mallick, K., Lee, J., 2012. The potential of potential evapotranspiration. American Geophysical Union, San Francisco, California, USA.

Also, we could send you our results and draft paper thus far, and you could draw inspiration and/or take whatever you want from it to help your paper out. We have a few other ways of looking at it. If so, you could add her as a co-author and make her

decade :)
* * *
Interactive
comment

---

## Referee Comment (RC1) · Anonymous Referee #1 · 2 Mar 2018

**General Comments**

This is a landmark paper. The authors introduce a wealth of hard-won, empirical data into a longstanding debate over how best to parameterize potential evapotranspiration (PET). Their analysis is meticulous, thorough, and well documented. Their main finding (robustness of Milly-Dunne and Priestley-Taylor methods, relative to Penman-Monteith method) is convincing and will surprise many investigators.

**Specific Comments**

My most significant concern with this analysis was the use of evaporative fraction, LE/(LE+H), in the characterization of stress. I wondered if this might somehow bias the analysis in favor of the Milly-Dunne method, since MD posits a constant value

of LE/(LE+H). For this reason, the sensitivity analysis using soil moisture as a stress criterion, and reaching similar conclusions, is a valuable part of the paper.

One other concern that might be allayed by a little more information is the use of "data corrected by energy balance closure." For one who is not familiar with FLUXNET and might hesitate to dig into the Michel et al 2016 reference, could the authors say just a little more about how this method works, how big the typical adjustments are, and to what extent the correction method could potentially influence the findings?

If the authors mean to suggest (this is not entirely clear, and should be clarified) that the simple radiation-based methods should now be incorporated into climate models, then I disagree. Climate models use the same physics upon which Penman-Monteith is based, but it needs to be recognized that in such models the stomatal conductance is calculated dynamically in response to controlling environmental variables. (There is nothing wrong with the Penman-Monteith approach in principle; it's just that it's hard to apply observationally, since it is sensitive to variables that are hard to know with sufficient precision.) Furthermore, the value of stomatal conductance is crucial for the computation of land-atmosphere carbon exchange. I would agree, on the other hand, that the MD method beats the PM method hands-down for application in global "offline" analyses in which atmospheric feedbacks are not present. However, it can also be argued that analysis of climate-model outputs themselves is a better way to spend one's time than doing offline analyses, which are sometimes amount to nothing better than attempts make a silk purse from a sow's ear.

To address the question of whether or not PET can be calculated correctly from actual Rn-G when the system is stressed takes this otherwise solid empirical paper into the metaphysical realm. What is the meaning of PET in a stressed system? And how does one empirically test that meaning? If one considers the feedback to surface temperature and albedo, why not also the feedback to lower atmospheric conditions, such as humidity and temperature, leading to changes in downward longwave radiation? This passage, for me, detracts from the paper and might better be presented as a technical

note elsewhere. Highlighting it in the Conclusion, at the expense of more concrete and surprising findings, seems not to be an ideal choice.

Could the authors compose a more fitting title? It is nice that they have provided estimates of PET all over the world, but the scientific value of the paper lies in its use of these data to test conceptual frameworks and related equations for quantifying PET.

Technical Corrections/Comments

P1L6 (i.e., Page 1, Line 6). "forecastING" P1,L14. "calibrated BY biome" (here and many places elsewhere through the paper) P2L39. "compared" P3L16. "atmospheric demand" seems an inappropriate phrase, given the dependence on surface properties. P5L14. "will be used, IN ADDITION TO a biome-specific" P7L5. "and WHERE u*" P8L17. "if FEWER than" P8L18. "criteriON" P8L26-27. I don't think the authors mean "actual crop" here but rather "actual vegetation" P9L34. Seems more significant than "marginal" to me. P10L7. alpha_RB typo? P10L17-18. Fig 3d rather than Fig 3c? P13L8. "SMOOTHS" P13L13. "relating to whether leaves" P13L17. "issues and would" P21. Thornthwaite is misspelled. Table 2. Use of color is a little distracting/unnecessary, and dark shades obscure text in first column. Why not use horizontal and vertical lines to serve same purpose? Table 2. I am not familiar with the a/b notation in the SUPERscripts (not subscripts). Is there a simple explanation so the reader doesn't need to search through a statistics book? Table 4,5,6. Again, reconsider use of color. Table 5,6. Could it be informative/helpful also to highlight the values that give the best results within the limits of the "standard" approach? Figure 1. The grey background is so dark that it reduces contrast with colored symbols. Could be lighter, or just put in coastlines. The symbols are very difficult to differentiate within a given color set. Figure 2. Label dates with full year, e.g., 2010. Figure 4. The symbols are very difficult to differentiate within a given color set. And really there is too much information on these plots, making it difficult to see the authors' main point. What about a 3x6 matrix of panels (some r-s-b spaces empty) with a single panel showing data from the six (or

other number of) example sites? Figure 6. I don't understand the dotted, dashed, and solid lines. This plot overall is hard to read any might possibly be improved upon.

---

## Referee Comment (RC2) · Anonymous Referee #2 · 4 Mar 2018

**Review of HESS manuscript # HESS_2017_682**

**Title: Potential evaporation at eddy-covariance sites across the globe**

This is a well-organized and comprehensive manuscript that uses the broad FLUXNET datasets to assess some important and well-known methods for estimation of potential evaporation. The findings here could provide a basis for hydrological and climatological analysis and modeling. Despite merits of the work, certain aspects need to be clarified to improve its impact and avoid confusion of readers.

- Although the authors have nicely reviewed different definitions and ambiguities relevant to "potential evaporation", there is no discussion in the context of Complementary Relationship (CR). Based on the CR, the potential evaporation serves as a dynamic measure of evaporative demand reflecting the land-atmosphere coupling as land dries; hence, what is provided in this work as the potential evaporation is more consistent with definition of wet surface evaporation or a reference evaporation where water availability is not limited (for example, see Brutsaert [2005], Kahler and Brutsaert [2006], and Aminzadeh, Or and Roderick [2016]). I am not also sure what should be the exact definition for potential evaporation, but still prefer to call what you considered as "reference evaporation" or "unstressed evaporation" rather than "potential evaporation" (somehow reflected in the last lines of section 2.4).

- The effect of scales has been discussed in page 2 (line 6) arguing that reference surface should not affect the meteorological condition, what about the effect of meteorological forcing on evaporation from that reference surface? Here is the place for discussion of feedbacks.

- Figure 2a: what is the reason for difference between dots and dark gray line? I understand they are calculated based on Eq. (9), but such difference between half-hourly and daily values is not intuitive! Looking at section 3.1 in Pennypacker and Baldocchi [2016], the daily VH is calculated from daily average friction velocity and drag coefficient and not aggregation of half-hourly VH values obtained from half-hourly database.

- The $PT_r$ and $PT_s$ are based on $\alpha_{PT} = 1.26$. Based on data in Table 4, we see there is a good performance for both (especially $PT_s$) regardless of the vegetation type. Considering that $\alpha_{PT} = 1.26$ was obtained from measurements (basically) over water

bodies (e.g., oceans), and noting that energy partitioning over a water body is quite different with land surfaces, what is the reason for such nice performance here?

- Page 4, line 5: I doubt even for a well-watered canopy $r_c=0$; this is nicely shown in Plate 1 of Baldocchi et el. [1997].

- Based on the criterion described in section 2.2 for aggregating sub-daily measurements, it is not clear what happened for cloudy days when surface shortwave incoming radiation is used instead if radiation at top of atmosphere.

- The discussion in page 2, line 17 is not consistent; I think the lower skin temperature yields a higher net radiation (less outgoing longwave radiation); please check.

- Although the main analysis of unstressed days is based on the energy criterion, the definition of unstressed days based on soil moisture is a bit questionable as an unstressed day is recognized based on $98^{th}$ percentile of measurements in each site; what if a site has always very low water moisture levels?

- Page 10, line 11: is there any specific reason/interpretation? Intuitively the wind speed would strongly affect turbulent transfer and, in turn, $E_{unstr}$.

- Table 1: why you need to calculate $r_{aH}$ for PT and MD?!

- Is $T_{eff}$ in Eq. (5) in degree of Celsius?

---

## Author Comment (AC1) · 26 Mar 2018

Dear Dr. Fisher,

thank you for pointing out this study. We will cite it in the upcoming revised version of the paper.

Kind regards,

Wouter Maes,

on behalf of the co-authors.

---

## Author Comment (AC2) · 26 Mar 2018

**Response to comments by Anonymous Referee #1**

**General Comments**

*This is a landmark paper. The authors introduce a wealth of hard-won, empirical data into a longstanding debate over how best to parameterize potential evapotranspiration (PET). Their analysis is meticulous, thorough, and well documented. Their main finding (robustness of Milly-Dunne and Priestley-Taylor methods, relative to Penman-Monteith method) is convincing and will surprise many investigators.*

*Specific Comments*

1) *My most significant concern with this analysis was the use of evaporative fraction, LE/(LE+H), in the characterization of stress. I wondered if this might somehow bias the analysis in favor of the Milly-Dunne method, since MD posits a constant value of LE/(LE+H). For this reason, the sensitivity analysis using soil moisture as a stress criterion, and reaching similar conclusions, is a valuable part of the paper.*

> Response: The reviewer is indeed raising a legitimate concern. For this reason we included the soil moisture criterion as well, which nicely confirms the outcome of the analysis based on the evaporative fraction. This will be further stressed in the revised manuscript by adding to the end of Section 4.1.: "*Still, by using the evaporative fraction as a criterion for selecting unstressed days, we might bias the findings in favour of the PT and MD methods, as they are more sensitive to the available energy. However, the soil moisture criterion taken here provides an independent check of the results and confirms the robust and superior performance of the PT$_b$ and MD$_b$ methods*".

2) *One other concern that might be allayed by a little more information is the use of "data corrected by energy balance closure." For one who is not familiar with FLUXNET and might hesitate to dig into the Michel et al 2016 reference, could the authors say just a little more about how this method works, how big the typical adjustments are, and to what extent the correction method could potentially influence the findings?*

> Response: We will add a section explaining this in the text: "*In this approach, the Bowen ratio derived from the tower measurements is assumed to be correct, and the measured λE$_a$ and H are multiplied by a correction factor derived from a moving window method; see http://fluxnet.fluxdata.org/data/fluxnet2015-dataset/data-processing/ for a detailed description.*" We tested both the use of the 'raw' and energy-balance-corrected values, and found very little differences in the results of both products.

3) *If the authors mean to suggest (this is not entirely clear, and should be clarified) that the simple radiation-based methods should now be incorporated into climate models, then I disagree. Climate models use the same physics upon which Penman-Monteith is based, but it needs to be recognized that in such models the stomatal conductance is calculated dynamically in response to controlling environmental variables. (There is nothing wrong with the Penman-Monteith approach in principle; it's just that it's hard to apply observationally, since it is sensitive to variables that are hard to know with sufficient precision.) Furthermore, the value of stomatal conductance is crucial for the computation of land-atmosphere carbon exchange. I would agree, on the other hand, that the MD method beats the PM method hands-down for application in global "offline" analyses in which atmospheric feedbacks are not present. However, it can also be argued that analysis of climate-model outputs themselves is a better way to spend one's time than doing offline analyses, which are sometimes amount to nothing better than attempts make a silk purse from a sow's ear.*

> Response:
> We fully agree with the reviewer on this matter. We do not want to suggest the incorporation in online climate models, which require solving for stomatal conductance to compute carbon fluxes,

and which do not rely on the calculation of potential evaporation to derive actual rates. We will make sure that in the revised version that that interpretation is not inferred. We do indeed recommend the inclusion of simple radiation-based methods in offline computations, such as drought monitoring systems of rainfall-runoff models. Also, we are in communication with the FLUXNET community to include the estimates of $E_p$ as part of the FLUXNET synthesis dataset.

4) *To address the question of whether or not PET can be calculated correctly from actual Rn-G when the system is stressed takes this otherwise solid empirical paper into the metaphysical realm. What is the meaning of PET in a stressed system? And how does one empirically test that meaning? If one considers the feedback to surface temperature and albedo, why not also the feedback to lower atmospheric conditions, such as humidity and temperature, leading to changes in downward longwave radiation? This passage, for me, detracts from the paper and might better be presented as a technical note elsewhere. Highlighting it in the Conclusion, at the expense of more concrete and surprising findings, seems not to be an ideal choice.*

    Response: We understand the reviewer's concern that this section deviates from the solid empirical evidence into a 'metaphysical realm'. However, the question on how to best calculate $E_p$ – also in unstressed conditions – is an important one, and this section discusses the best ways to do it. Therefore, we decided to leave it into the Discussion section, yet to exclude it from the Conclusions. While it is clear that PET is also relevant in a stressed system – even more than in an unstressed system –it should also be clear that "*it is nearly impossible to define a correct and universally accepted definition of $E_p$, and the most appropriate definition should remain tied to the specific interest and application*", as mentioned in the introduction section. We understand that the correction of surface variables but not considering the feedback to near-surface atmospheric conditions is somewhat arbitrary. We wanted to draw the line at the surface and consider only the actual forcing variables as 'input' and not aim to correct for feedbacks into the radiation, which is extremely difficult and impractical to estimate reliably – as mentioned in a new section in the introduction discussing feedbacks "*Moreover, extensive reference surfaces can be expected to not only exert a feedback on the aerodynamic forcing, but also on the radiative forcing. Indeed, by altering the temperature, humidity and through cloud formation, extensive reference systems are likely to also affect incoming shortwave and longwave radiation. Yet, as this feedback is almost impossible to calculate, it is ignored in all methods considering extensive reference surfaces*". After all, that is the only way to still relay on real tower measurements of atmospheric forcing during stress times. We can certainly see the need to correct for the more direct effect of soil moisture on G, $SW_{out}$ and $LW_{out}$, because they are more immediately influenced and thus different between stressed and unstressed ecosystems. Therefore, we prefer not to attempt to correct for near-surface atmospheric variables.

    We also want to highlight that we also show that: (a) the simple alternative of ignoring this issue, and taking the actual ($R_n$–G) as forcing variable for calculating $E_p$, leads to a severe underestimation of $E_p$; (b) results suggest that this underestimation is largely caused by differences in $LW_{out}$ between the stressed and the unstressed ecosystem; (c) there is a practical solution to overcome this underestimation, which only requires $T_a$ as additional input, and which results in an almost unbiased estimate of $E_p$.

    Finally, we will also acknowledge the complementary approach in our revised paper along these lines, after the suggestion by Reviewer #2.

5) *Could the authors compose a more fitting title? It is nice that they have provided estimates of PET all over the world, but the scientific value of the paper lies in its use of these data to test conceptual frameworks and related equations for quantifying PET.*

    Response: We understand the rationale behind the comment, however, we do consider the title to be well suited for the article. In this publication, we are "targeting" different scientific communities, and probably, the scientific value for each community will be different. The current title highlights the end product (dataset of $E_p$), which can be of great interest for e.g. the FLUXNET community. A title that is

more fitted to the findings, such as "Radiation-based methods are more optimal way to estimate potential evaporation at ecosystem scale", would indeed be more appealing to the hydrological modelling community, but we believe that will narrow down the potential audience of the manuscript.

*Technical Corrections/Comments*
*P1L6 (i.e., Page 1, Line 6). "forecastING"*
   Thanks. We will correct it.
*P1,L14. "calibrated BY biome" (here and many places elsewhere through the paper)*
   Thanks, this will be changed throughout the paper.
*P2L39. "compared"*
   True.
*P3L16. "atmospheric demand" seems an inappropriate phrase, given the dependence on surface properties.*
   'atmospheric' was omitted.
*P5L14. "will be used, IN ADDITION TO a biome-specific"*
   We will change it.
*P7L5. "and WHERE u\*"*
   Will be changed.
*P8L17. "if FEWER than".*
   True, thanks.
*P8L18. "criteriON".*
   It will be corrected.
*P8L26-27. I don't think the authors mean "actual crop" here but rather "actual vegetation"*
   Correct, and corrected accordingly
*P9L34. Seems more significant than "marginal" to me.*
   The statement "*although these differences are only slightly significant in the case of $g_{c\_ref}$ (p=0.017 – see Table 2)*" will be removed.
 *P10L7. alpha_RB typo?*
   Yes, corrected to $\alpha_{MD}$.
*P10L17-18. Fig 3d rather than Fig 3c?*
   Indeed, adjusted.
*P13L8. "SMOOTHS".*
   True*.*
*P13L13. "relating to whether leaves".*
   It will be changed.
*P13L17. "issues and would".*
   It will be corrected.
*P21. Thornthwaite is misspelled.*
   Thanks, corrected

*Table 2. Use of color is a little distracting/ unnecessary, and dark shades obscure text in first column. Why not use horizontal and vertical lines to serve same purpose?*
  Response: The colours were used to group biomes into forest/savannah /grassland/crop/wetland ecosystems.  This will be specified. The darker colours will be adjusted.

*Table 2. I am not familiar with the a/b notation in the SUPERscripts (not subscripts). Is there a simple explanation so the reader doesn't need to search through a statistics book?*
  Response: We explained it in the new version as "*Different alphabetic superscripts indicate significantly differing means (Tukey post-hoc test; p<0.05).*"

*Table 4,5,6. Again, reconsider use of color.*
  Response: *See response on Table 2.*
*Table 5,6. Could it be informative/helpful also to highlight the values that give the best results within the limits of the "standard" approach?*

Response: We tried this, but it results in a messy figure. However, since the best biome-specific approach can be calculated for nearly the entire world based on the datasets, there is arguably no need to revert to the standard approaches.

*Figure 1. The grey background is so dark that it reduces contrast with colored symbols. Could be lighter, or just put in coastlines. The symbols are very difficult to differentiate within a given color set.*

Response: Figure 1 has been redone – the grey background is now lighter, and symbols were changed to add clarity to the figure:

[Figure]

**Updated version of** **Figure 1. Location of the flux sites used in this study per biome. CRO=cropland; DBF=Deciduous Broadleaf Forest; EBF=Evergreen Broadleaf Forest; ENF=Evergreen Needle Forest; MF=Mixed Forest; CSH=Closed Shrubland; WSA=Woody Savannah; SAV=Savannah; OSH=Open Shrubland; GRA=Grasslands; WET=Wetlands.**

*Figure 2. Label dates with full year, e.g., 2010.*
      Response: Will be done as requested:

[Figure]

*Updated version of* **Figure 2 (a) Vegetation height dynamics in time (grey dots: half-hourly measurements; dark grey lines: daily mean vegetation height; red line: 30-day moving average (i.e. the final vegetation height dataset). (b) Relation between the Stanton number ($k$B$^{-1}$) and the Reynolds number (Re). Both plots correspond to the woody savannah site of Santa Rita Mesquite (Arizona, USA).**

*Figure 4. The symbols are very difficult to differentiate within a given color set. And really there is too much information on these plots, making it difficult to see the authors' main point. What about a 3x6 matrix of panels (some r-s-b spaces empty) with a single panel showing data from the six (or other number of) example sites?*

Response: We understand the comment, we will improve the figures as requested, using 6 different panels per site and showing three sites (showing 18 panels would only allow showing 1 site):

[Figure]

*Updated version of* **Figure 4. Scatterplot of the measured $E_{unstr}$ versus $E_p$ calculated with the different methods for three selected sites. The discontinuous line is the 1:1 line. Based on unstressed days only defined using the energy balance criterion.**

*Figure 6. I don't understand the dotted, dashed, and solid lines. This plot overall is hard to read and might possibly be improved upon.*

Response: We understand the reviewer's comments and will improve the figure as follows:

[Figure]

**Updated version of Figure 6. Distribution of the mean error per fluxtower in the estimate of $E_p$ of two empirical methods to calculate unstressed $(R_n\text{-}G)$. The first empirical method simply take the actual $(R_n\text{-}G)$ as input, the second method corrects the actual $(R_n\text{-}G)$ with $T_a$ (Eq. (16)). Negative Y-values indicate an underestimation by the empirical methods. For each distribution, the mean and median are indicated with a full and dashed line, respectively.**

---

## Author Comment (AC3) · 26 Mar 2018

**Response to comments by Anonymous Referee #2**

Hey guys, we actually did this study back in 2012, presented at AGU in a session
chaired by Martha Anderson, but we never ended up publishing it. No worries though–
we snooze we lose! But, it would be nice if you wouldn't mind citing that presentation?
It was led by a high-school student, so I think she'd be thrilled to be cited.
Palmer, C., Fisher, J.B., Mallick, K., Lee, J., 2012. The potential of potential evapotranspiration.
American Geophysical Union, San Francisco, California, USA.
Also, we could send you our results and draft paper thus far, and you could draw
inspiration and/or take whatever you want from it to help your paper out. We have a
few other ways of looking at it. If so, you could add her as a co-author and make her

*Title: Potential evaporation at eddy-covariance sites across the globe*

*This is a well-organized and comprehensive manuscript that uses the broad FLUXNET datasets to assess some important and well-known methods for estimation of potential evaporation. The findings here could provide a basis for hydrological and climatological analysis and modeling. Despite merits of the work, certain aspects need to be clarified to improve its impact and avoid confusion of readers.*

1) *Although the authors have nicely reviewed different definitions and ambiguities relevant to "potential evaporation", there is no discussion in the context of Complementary Relationship (CR). Based on the CR, the potential evaporation serves as a dynamic measure of evaporative demand reflecting the land-atmosphere coupling as land dries; hence, what is provided in this work as the potential evaporation is more consistent with definition of wet surface evaporation or a reference evaporation where water availability is not limited (for example, see Brutsaert [2005], Kahler and Brutsaert [2006], and Aminzadeh, Or and Roderick [2016]). I am not also sure what should be the exact definition for potential evaporation, but still prefer to call what you considered as "reference evaporation" or "unstressed evaporation" rather than "potential evaporation" (somehow reflected in the last lines of section 2.4).*

   Response: Reference to the CR was indeed absent in the previous version and we agree with the referee that it deserves to be incorporated in the paper. We will acknowledge the CR in the Introduction and Discussions section. We also agree that our definition of potential evaporation as "unstressed" evaporation closely matches the $E_{p0}$ (well-watered surface) in the CR definition, and this will now be mentioned in the text as well. ("*Based on the above review, $E_p$ is defined using the actual ecosystem evaporating at maximal rate as reference system, so $E_p$ refers to the actual demand for water experienced by the ecosystem. This definition is similar to that of $E_{p0}$ in the complementary relationship.*").

   However, we do not necessarily agree that the terminology should be changed in the manuscript. We realise that in the CR, the term 'potential evaporation' is used mainly for the 'real' potential evaporation ($E_{pa}$ in our paper) and, to add confusion, not for the 'free' potential evaporation ($E_{p0}$ or $E_w$). However, the term 'potential evaporation' is widely used by the climate community, often with the same meaning as we uptake here. Without denying the controversy surrounding the term, but acknowledging it, we understand we should be very precise on how we define potential evaporation in the paper. We believe that the inclusion of the CR will add more context to this definition and make things clearer for all readers.

2) *The effect of scales has been discussed in page 2 (line 6) arguing that reference surface should not affect the meteorological condition, what about the effect of meteorological forcing on evaporation from that reference surface? Here is the place for discussion of feedbacks.*

   Response: We agree. In the next version, we will discuss potential feedback mechanisms more clearly. We will also incorporate this to the Discussion section, as suggested by Referee #1.

3) *Figure 2a: what is the reason for difference between dots and dark gray line? I understand they are calculated based on Eq. (9), but such difference between half-hourly and daily values is not intuitive!*

*Looking at section 3.1 in Pennypacker and Baldocchi [2016], the daily VH is calculated from daily average friction velocity and drag coefficient and not aggregation of half-hourly VH values obtained from half-hourly database.*

Response: - We believe that Pennypacker and Baldocchi (2016) did not use daily average friction velocity and drag coefficients, but averaged out height data from half-hourly to daily averages. This is also the approach we used, but we further smoothed the data using a moving window. The apparent difference between half-hourly and daily values in Fig. 2a is indeed misleading – the large majority of the half-hourly height estimates are close to the daily average – this would better be visualised with a density graph. Because we understand the reviewer's concern about this figure and agree that showing the half-hourly data in the format it was presented can be confusing, in the new version we will leave out the half-hourly dots.

[Figure]

*Updated version of* **Figure 2 (a) Vegetation height dynamics in time (grey dots: half-hourly measurements; dark grey lines: daily mean vegetation height; red line: 30-day moving average (i.e. the final vegetation height dataset). (b) Relation between the Stanton number ($k$B$^{-1}$) and the Reynolds number (Re). Both plots correspond to the woody savannah site of Santa Rita Mesquite (Arizona, USA).**

4) *The $PT_r$ and $PT_s$ are based on $\alpha_{PT}$=1.26. Based on data in Table 4, we see there is a good performance for both (especially $PT_s$) regardless of the vegetation type. Considering that was obtained from measurements (basically) over water bodies (e.g., oceans), and noting that energy partitioning over a water body is quite different with land surfaces, what is the reason for such nice performance here?*

Response: In Table 4, only correlations are considered,. Because the $\alpha_{PT}$ value here is a multiplying factor of $\frac{s\,(R_n-G)}{s+\gamma}$, the value of $\alpha_{PT}$ as such does not have an effect on the correlations. This is why $PT_s$ and $PT_B$ have the same correlation, as do $MD_s$ and $MD_B$ as well as $Ou_s$ and $Ou_B$. On the other hand, the relatively high unbiased RMSE and bias $PT_r$ and $PT_s$ (see Tables 5 and 6) indicate that assuming a value of $\alpha_{PT}$=1.26 introduces and overestimation.

5) *Page 4, line 5: I doubt even for a well-watered canopy $r_c$=0; this is nicely shown in Plate 1 of Baldocchi et el. [1997].*

Response: We believe there is some confusion between a well-watered and a wet canopy. Indeed, as correctly observed by the referee, even for a well-watered canopy, $r_c$ > 0. Only for a wet canopy surface

(e.g. shortly after rainfall, ...), $r_c=0$. This is also considered in traditional interception loss models (see e.g. Rutter et al., 1975, DOI: 10.2307/2401739).

6) *Based on the criterion described in section 2.2 for aggregating sub-daily measurements, it is not clear what happened for cloudy days when surface shortwave incoming radiation is used instead if radiation at top of atmosphere.*

Response: We used the same threshold; this is now specified in the text. Note that a (half-)hourly mean of 5 Wm$^{-2}$ is a very low light intensity, so even under very cloudy conditions, the use of SW$_{in}$ does not hold the risk of excluding data in the middle of the day. TOA-atmosphere is almost always available for all sites (as it can be calculated directly).

7) *The discussion in page 2, line 17 is not consistent; I think the lower skin temperature yields a higher net radiation (less outgoing longwave radiation); please check.*

Response: Thank you, indeed. This will be corrected in the text.

8) *Although the main analysis of unstressed days is based on the energy criterion, the definition of unstressed days based on soil moisture is a bit questionable as an unstressed day is recognized based on 98th percentile of measurements in each site; what if a site has always very low water moisture levels?*

Response: The reviewer touches upon an assumption that needs to be made with any method extracting a subset of unstressed days from the time series of measured variables: that the dataset of each site contains at least a few dates in which unstressed conditions prevail. This is an assumption that is made for both the soil moisture and the energy balance criterion. However, there are not true arid sites in the database for which one could not assume that the conditions are unstressed for a few days per year. Even in the case of e.g. Au-TTE and Au-ASM (both very dry sites in the centre of Australia), there is a wet season clearly reflected in the observations. H

To further clarify, in the case of the soil moisture criterion, we divided the dataset into 20$^{th}$ percentiles of evaporation, and selected within all but the lowest percentile the days with 5% highest soil moisture. Specifically to avoid selecting days in which soil moisture is relatively high but still limiting, we added a second condition: the soil moisture of these days needed to be above 75% of the maximum soil moisture of the site, else, these days were removed from the unstressed dataset. This maximum soil moisture was defined as the 98$^{th}$ percentile of soil moisture (we took the 98$^{th}$ percentile, rather than the absolute measured maximum, to avoid influence from extreme values).

9) *Page 10, line 11: is there any specific reason/interpretation? Intuitively the wind speed would strongly affect turbulent transfer and, in turn, $E_{unstr}$.*

Response: Indeed, intuitively, while one would expect wind speed to have a strong effect on $E_{unstr}$, this is not shown in our study. However, this finding is quite consistent with the observations throughout our study: $E_{unstr}$ is predominantly determined by radiation ($R_n$) (see left column of Table 3) and can best be estimated with the Milly and Dunne method, in which wind speed is not considered.

10) *Table 1: why you need to calculate $r_{aH}$ for PT and MD?!*

Response: Thanks for pointing this out, this was indeed an error – we will correct Table 1. We noted a similar issue with RH/VPD, which will also be corrected.

11) *Is $T_{eff}$ in Eq. (5) in degree of Celsius?*
Response: yes it is, see e.g. Pereira and Pruitt, 2004 (doi:10.1016/j.agwat.2003.11.003)

---

## Referee Comment (RC3) · Anonymous Referee #3 · 7 May 2018

The manuscript titled 'Potential evaporation at eddy-covariance sites across the globe' is a surprising piece of work to read through. The reasons are as follows:

(1) The title of the paper is inappropriate in my view. What the authors have done is they selected the events of unlimited soil moisture and/or high EF events and used a host of predictive Potential evaporation models to calculate the statistical errors of the models, based on which the appropriateness of the models are highlighted. The title of the paper should be 'Unstressed evaporation modeling at eddy covariance sites across the globe'.

(2) It is obvious that under unlimited soil moisture, radiation explained maximum variability in evaporation. Such results have been published in many literatures and not

new to the community. However, flagging it as potential evaporation is misleading. It should be seen as actual evaporation under unlimited soil moisture which is driven by radiation only. The authors should realize that potential evaporation is a notional term.? What happens in desert where high radiation load is accompanied by extremely high VPD? If we plot an image of global Ep distribution, we will see the deserts to have the maximum Ep values. Then how would one can pick potential evaporation events based on EF or soil moisture saturation. Although the authors have hinted (in Page 3, L15) that Ep is the potential evaporative demand, but finally inclined to wettest events instead of looking at the evaporative demand.

(3) The estimation rAH is extremely outdated and the no attempt is made to demonstrate how sensitive is the PM and Penman equation to rAH parameterization, which in my opinion should carry a section of results, instead of concluding biome specific PT is consistently better that any other models. Yes, when the evaporation is driven by radiation only, it is no wonder that PT will do a good job. However the calibration of PT was still needed to adjust the hidden VPD and rAH related variability in the 'alpha' parameter.

(4) How the residual ET error in PM and Penman was related to rAH? It is now becoming prominent to the ET community that rAH parameterizations are ambiguous and this needs to be resolved in surface energy balance modelling. Some recent studies have highlighted the importance of analytical estimation of aerodynamic condurtance to overcome the uncertainties in ET modelling, which authors are expected to be aware of.

(5) What about the feedback that rAH provides to the evaporation? Without considering those feedbacks, it would be unjustified to come to conclusion bout PM or Penman equations (as mentioned in section 4.3).

(6) Estimation of gc_ref is purely climatological and as a result the differences in gc_ref between the biomes are marginally significant.

(7) A Table of symbols and variables would be very helpful to the readers.

(8) section 4.3: A complete change of description is needed. The conclusion of Michel et al., 2016 and Ershadi et al., 2014 was an outcome of outdated conductance parameterization (despite they were published) and should not be used to as a justification in the discussion.

(9) section 4.3: It is important to highlight the fact that the conductances (both gAH and gc) in the PM equation provides feedback in evaporation that changes the aerodynamic vapor pressure and temperatures. This study used empirical gAH model to obtain evaporation estimates from PM and Penman. In addition the authors made an effort to show gC-VPD known curve to justify the results. In the present case, justification on why PM and Penman equation is complex should come from analysis of gAH and linking the model errors with empirical uncertainties in gAH.

(10) Also, the authots did not mention if they took care of the sky conditions. Ideally the study should select clear sky cases.

Finally, I would like to thank the authors for the honest effort to use large fluxnet dataset and untap the events of unstressed evaporation. But this should not be seen as potential evaporation. A detailed analysis of the role of gAH in PM, additional role of VPD in creating the differences in evaporation between PTb, PM, and Penman would make the study worthy of publication.

---

## Author Comment (AC4) · 7 May 2018

Dear Anonymous referee, editor,

We are glad to see this discussion being reopened despite the delay, and we appreciate this new criticism. Please allow us to reply to these comments below (see blue fonts).

The manuscript titled 'Potential evaporation at eddy-covariance sites across the globe' is a surprising piece of work to read through. The reasons are as follows:

(1) The title of the paper is inappropriate in my view. What the authors have done is they selected the events of unlimited soil moisture and/or high EF events and used a host of predictive Potential evaporation models to calculate the statistical errors of the models, based on which the appropriateness of the models are highlighted. The title of the paper should be 'Unstressed evaporation modeling at eddy covariance sites across the globe'.
There are multiple definitions of potential evaporation ($E_p$) in literature. The evaporation occurring under (hypothetical or actual) unstressed conditions is in fact one of them. This is the definition of $E_p$ we uptake here, as the reviewer mentions, and as it is clearly stated in the manuscript. This agrees with multiple previous articles – just to cite some of them, see Douglas *et al.* (2009), Pereira and Pruitt (2004), Katerji and Rana (2011), Li *et al.* (2016), Jacobs *et al.* (2004), Fisher *et al.* (2011).

(2) It is obvious that under unlimited soil moisture, radiation explained maximum variability in evaporation. Such results have been published in many literatures and not new to the community.
We appreciate the comment. Indeed, there are plenty of articles that sustain that radiation is the main driver of evaporation under potential rates. However, while for the referee it appears to be an obviosity, many authors – like e.g. Thornthwaite or Odin – have proposed methods based on a rationale that contradicts this expectation. While we believe that it is important to highlight the role of radiation as dominant driver, we certainly do not claim this is a novel result: we cite plenty of articles that agree with this finding in the 'References' section. If we have missed any relevant one, we invite the reviewer to suggest their preferred ones and we will incorporate them to the revised version.

However, flagging it as potential evaporation is misleading.
We disagree. See first response.

It should be seen as actual evaporation under unlimited soil moisture which is driven by radiation only.
The fact that $E_p$ is driven by radiation (not 'only', but mainly) is concluded from our findings that radiation-driven formulations perform better at estimating $E_p$ when it happens. And yes, for us, 'actual evaporation under no stress' is in fact $E_p$. We refer to the above-mentioned articles again and our chosen definition of $E_p$ stated in the first response.

The authors should realize that potential evaporation is a notional term?
Yes, we should... and in fact we do. We certainly understand it is a notional or idealised concept, as it is extensively discussed in the text. Our estimates of $E_p$ in days with stressed conditions correspond to the evaporation that would take place under no stress, bearing in mind the uncertainty due to possible feedbacks as extensively discussed. Again, if the reviewer finds incomplete any of the discussions we invite them to clarify what exactly should be expanded or amended.

What happens in desert where high radiation load is accompanied by extremely high VPD?
Evaporation under sufficient soil water availability would be larger for the same net radiation if VPD were lower. The reviewer certainly knows this... Maybe we are misinterpreting the question.

If we plot an image of global Ep distribution, we will see the deserts to have the maximum Ep values.

This depends again on the method used to estimate $E_p$, which happens to be a notional term. The referee is probably not acknowledging that the high albedo and land surface temperature in the deserts lead to a reduced net radiation, despite the high VPD. The result is that the $E_p$ estimated via radiation-driven approaches will show remarkably low values in the desert. This may come across as surprising to the reviewer, so we refer to e.g. Fisher *et al.* (2011) Figure 3. This is to highlight again the differences among the definitions – and subsequent formulations – of $E_p$ in current research. Nonetheless, as the reviewer also knows, there are no desert sites in the database: we did not apply the method to any arid climate (where the value of accurate potential transpiration estimates has arguably no value anyways).

Then how would one can pick potential evaporation events based on EF or soil moisture saturation.

See again our first response for our definition of $E_p$, and Section 2.4 in the article, and the literature referred here. This is common practice and is consistent with the definition of potential evaporation considered here.

Although the authors have hinted (in Page 3, L15) that Ep is the potential evaporative demand, but finally inclined to wettest events instead of looking at the evaporative demand.

We are interested in providing estimates of $E_p$ for wet and dry times. Our use of 'evaporative demand' here is as synonym to $E_p$, and consistent with our definition: the evaporation that would occur under unstressed conditions. We do not mean vapour pressure deficit, if this is what the referee means.

(3) The estimation rAH is extremely outdated and the no attempt is made to demonstrate how sensitive is the PM and Penman equation to rAH parameterization, which in my opinion should carry a section of results, instead of concluding biome specific PT is consistently better that any other models.

We strongly disagree with the reviewer if it is implied that the logarithmic wind profile method and the Monin-Obukhov theory are 'extremely outdated'. To date, these are the standard and optimal methods for calculating aerodynamic resistance ($r_{aH}$). In fact, we incorporated several aspects in the paper to ensure our calculation of $r_{aH}$ is the best available, specifically to avoid Penman methods to be disadvantaged. We estimated the vegetation height using a recent new method by Pennypacker and Baldocchi (2016), incorporated stability functions using the best available methods (Garratt, 1992; Brutsaert, 2005), and incorporated a parameterisation of the Stanton number based on recent insights by Li *et al.* (2017). Prior to submission, this paper was sent to all the principal investigators of all 107 flux towers, many of which are very well acquainted with the difficulties of estimating $r_{aH}$. Many of them replied and none of them had a complain about this methodology.

Of course, we would still appreciate it if the reviewer pointed us to the newer formulations that they are referring to. This would make this comment (and others!), in fact, constructive and helpful. Meanwhile, we trust the references highlighting that our $r_{aH}$ calculation is the state-of-the art. See from the last couple of months only: Valayamkunnath *et al.* (2018), Lin *et al.* (2018), Nyman *et al.* (2018), Srivastava *et al.* (2018), Yan *et al.* (2018).

Yes, when the evaporation is driven by radiation only, it is no wonder that PT will do a good job. However the calibration of PT was still needed to adjust the hidden VPD and rAH related variability in the 'alpha' parameter.

Absolutely right. We are glad the referee agrees with the discussion in the article! Then, of course if radiation was not the main driver, with a constant alpha per biome type PT would not outperform more complex methods.

(4) How the residual ET error in PM and Penman was related to rAH? It is now becoming prominent to the ET community that rAH parameterizations are ambiguous and this needs to be resolved in surface energy balance

modelling. Some recent studies have highlighted the importance of analytical estimation of aerodynamic condurtance to overcome the uncertainties in ET modelling, which authors are expected to be aware of.

Unfortunately, the reviewer failed again to add the references to these articles. We therefore choose to believe the findings from the above-mentioned recent literature.

(5) What about the feedback that rAH provides to the evaporation? Without considering those feedbacks, it would be unjustified to come to conclusion bout PM or Penman equations (as mentioned in section 4.3).

As mentioned in the response documents addressing the constructive feedback from previous referees, the consideration of atmospheric feedbacks will be further discussed in the revised manuscript.

(6) Estimation of gc_ref is purely climatological and as a result the differences in gc_ref between the biomes are marginally significant.

The estimation of $g_{c\,ref}$ is done by considering it as the residual term in the PM equation. It is therefore not purely climatological but reflects e.g. ecosystem roughness, species-based genetic differences in maximum stomatal conductance, etc. We believe this is clearly discussed in the article but will be further clarified in the revised version.

(7) A Table of symbols and variables would be very helpful to the readers.

This is indeed a constructive comment. We will incorporate it to the revised version.

(8) section 4.3: A complete change of description is needed. The conclusion of Michel *et al*, 2016 and Ershadi *et al*, 2014 was an outcome of outdated conductance parameterization (despite they were published) and should not be used to as a justification in the discussion.

See above response. To our understanding, both articles mentioned here were highly welcomed by the scientific community and have been highly cited. Could the reviewer perhaps share with us the list of articles he/she considers as 'state-of-the art', and the one containing the articles that should be blacklisted? That, together with a description of the reason we should not believe those articles, would make all these comments constructive.

(9) section 4.3: It is important to highlight the fact that the conductances (both gAH and gc) in the PM equation provides feedback in evaporation that changes the aerodynamic vapor pressure and temperatures. This study used empirical gAH model to obtain evaporation estimates from PM and Penman. In addition the authors made an effort to show gC-VPD known curve to justify the results. In the present case, justification on why PM and Penman equation is complex should come from analysis of gAH and linking the model errors with empirical uncertainties in gAH.

In section 4.3, we demonstrate that the flaw of using the Penman-Monteith based approaches for estimating $E_p$ is the assumption in these approaches that $g_{c\,ref}$ should be constant under unstressed surface conditions. This is not the case, and is the main reason why the PM approaches perform poorly when aiming to calculate $E_p$. We do not claim that the Penman-Monteith method as such is inferior in any way, yet the highly dynamic nature of the aerodynamic and surface conductance makes it difficult to apply reliably. As mentioned above, we tried to provide the best available method to estimate $r_{aH}$, and invite the reviewer to provide any improvement. However, this would not affect the underlying problem of the constant $g_{c\,ref}$ of the PM-based $E_p$ methods.

(10) Also, the authots did not mention if they took care of the sky conditions. Ideally the study should select clear sky cases.

We would like to sample all-sky conditions, since $E_p$ also matters when it is cloudy.

Finally, I would like to thank the authors for the honest effort to use large fluxnet dataset and untap the events of unstressed evaporation. But this should not be seen as potential evaporation. A detailed analysis of the role of

gAH in PM, additional role of VPD in creating the differences in evaporation between PTb, PM, and Penman would make the study worthy of publication.

See all responses above. Again, pointing us to the mentioned literature would make this a much more valuable exercise.

**References**

Brutsaert, W.: Hydrology: an introduction, Cambridge University Press, New York, 589 pp., 2005.

Douglas, E. M., Jacobs, J. M., Sumner, D. M., and Ray, R. L.: A comparison of models for estimating potential evapotranspiration for Florida land cover types, Journal of Hydrology, 373, 366–376, http://dx.doi.org/10.1016/j.jhydrol.2009.04.029, 2009.

Fisher, J. B., Whittaker, R. J. & Malhi, Y. ET come home: potential evapotranspiration in geographical ecology, Global Ecol Biogeography 20, 1–18 (2010).

Garratt, J. R.: The atmospheric boundary layer, Cambridge ; New York : Cambridge University Press, 1992.

Jacobs, J. M., Anderson, M. C., Friess, L. C., and Diak, G. R.: Solar radiation, longwave radiation and emergent wetland evapotranspiration estimates from satellite data in Florida, USA Hydrological Sciences Journal, 49, null-476, 10.1623/hysj.49.3.461.54352, 2004.

Katerji, N., and Rana, G.: Crop reference evapotranspiration: a discussion of the concept, analysis of the process and validation, Water Resources Management, 25, 1581–1600, 10.1007/s11269-010-9762-1, 2011.

Li, S., Kang, S., Zhang, L., Zhang, J., Du, T., Tong, L., and Ding, R.: Evaluation of six potential evapotranspiration models for estimating crop potential and actual evapotranspiration in arid regions, Journal of Hydrology, 543, Part B, 450–461, http://dx.doi.org/10.1016/j.jhydrol.2016.10.022, 2016.

Li, D., Rigden, A., Salvucci, G., and Liu, H.: Reconciling the Reynolds number dependence of scalar roughness length and laminar resistance, Geophysical Research Letters, 44, 3193-3200, 10.1002/2017GL072864, 2017.

Lin, C., Gentine, P., Huang, Y., Guan, K., Kimm, H., and Zhou, S.: Diel ecosystem conductance response to vapor pressure deficit is suboptimal and independent of soil moisture, Agricultural and Forest Meteorology, 250-251, 24-34, https://doi.org/10.1016/j.agrformet.2017.12.078, 2018.

Nyman, P., Baillie, C. C., Duff, T. J., and Sheridan, G. J.: Eco-hydrological controls on microclimate and surface fuel evaporation in complex terrain, Agricultural and Forest Meteorology, 252, 49-61, https://doi.org/10.1016/j.agrformet.2017.12.255, 2018.

Pennypacker, S., and Baldocchi, D.: Seeing the Fields and Forests: Application of Surface-Layer Theory and Flux-Tower Data to Calculating Vegetation Canopy Height, Boundary-Layer Meteorology, 158, 165-182, 10.1007/s10546-015-0090-0, 2016.

Pereira, A. R., and Pruitt, W. O.: Adaptation of the Thornthwaite scheme for estimating daily reference evapotranspiration, Agricultural Water Management, 66, 251–257, https://doi.org/10.1016/j.agwat.2003.11.003, 2004.

Srivastava, R. K., Panda, R. K., Chakraborty, A., and Halder, D.: Comparison of actual evapotranspiration of irrigated maize in a sub-humid region using four different canopy resistance based approaches, Agricultural Water Management, 202, 156-165, https://doi.org/10.1016/j.agwat.2018.02.021, 2018.

Valayamkunnath, P., Sridhar, V., Zhao, W., and Allen, R. G.: Intercomparison of surface energy fluxes, soil moisture, and evapotranspiration from eddy covariance, large-aperture scintillometer, and modeling across three ecosystems in a semiarid climate, Agricultural and Forest Meteorology, 248, 22-47, https://doi.org/10.1016/j.agrformet.2017.08.025, 2018.

Yan, H., Zhang, C., and Hiroki, O.: Parameterization of canopy resistance for modeling the energy partitioning of a paddy rice field, Paddy and Water Environment, 16, 109-123, 10.1007/s10333-017-0620-0, 2018.